EMBO
Molecular Medicine

# Restoration of defective oxidative phosphorylation to a subset of neurons prevents mitochondrial encephalopathy

Brittni R Walker [ID] [1], Lise-Michelle Theard[2], Milena Pinto[2,5], Monica Rodriguez-Silva[2], Sandra R Bacman[2] & Carlos T Moraes [ID] [2,3,4 ✉]

## Abstract

**Oxidative Phosphorylation (OXPHOS) defects can cause severe encephalopathies and no effective treatment exists for these disorders. To assess the ability of gene replacement to prevent disease progression, we subjected two different CNS-deficient mouse models (*Ndufs3*/complex I or *Cox10*/complex IV conditional knockouts) to gene therapy. We used retro-orbitally injected AAV-PHP.eB to deliver the missing gene to the CNS of these mice. In both cases, we observed survival extension from 5–6 to more than 15 months, with no detectable disease phenotypes. Likewise, molecular and cellular phenotypes were mostly recovered in the treated mice. Surprisingly, these remarkable phenotypic improvements were achieved with only ~30% of neurons expressing the transgene from the AAV-PHP.eB vector in the conditions used. These findings suggest that neurons lacking OXPHOS are protected by the surrounding neuronal environment and that partial compensation for neuronal OXPHOS loss can have disproportionately positive effects.**

**Keywords** Mitochondria; Gene Therapy; Mitochondrial Disease; Oxidative Phosphorylation
**Subject Categories** Genetics, Gene Therapy & Genetic Disease; Neuroscience; Organelles

## Introduction

Mitochondrial diseases are group of genetic diseases caused by mutations in nuclear DNA or mitochondrial DNA (mtDNA), ultimately affecting the oxidative phosphorylation (OXPHOS) system (Gorman et al, 2016). These diseases are often multisystemic, but tissues with high energy demand, such as muscle and the CNS are frequently affected (Gorman et al, 2016; Stewart and Chinnery, 2015).

The OXPHOS system is composed of five multi-subunits complexes. While portrayed as a linear pathway, the individual complexes associate with one another to form mega structures known as "OXPHOS supercomplexes" (Schäfer et al, 2006). Among the patient population, defects in complexes I and IV are prevalent (DiMauro et al, 2012; Distelmaier et al, 2009; Rodenburg, 2016). Complex I (CI) is the largest complex of the respiratory chain and is responsible for the electron transfer to ubiquinone from the oxidation of NADH as well as contributes to the proton gradient required for ATP synthesis (Wirth et al, 2016). It has 7 mtDNA-encoded and close to 40 nDNA-encoded (Wirth et al, 2016) subunits. Approximately a third of mitochondrial diseases stem from mutations and deficiencies in complex I and its activity (Distelmaier et al, 2009; Kirby et al, 1999; Rodenburg, 2016; Wirth et al, 2016). Mutations have been identified in all 7 of the mtDNA-encoded subunits, as well as 25 of the 37 nDNA-encoded subunits and 13 assembly factors. Clinical manifestations of CI deficiencies include Leigh Syndrome, ataxia, developmental delays, and ophthalmological abnormalities (Benit, 2004; Fassone and Rahman, 2012; Lake et al, 2016; Lou et al, 2018a, b).

Complex IV (CIV), also known as cytochrome c oxidase, is the terminal enzyme of the electron transport chain that catalyzes the transfer of electrons from reduced cytochrome c to molecular oxygen (Capaldi, 1990). It has 3 mtDNA-encoded subunits and 10 nDNA-encoded ones (Watson and McStay, 2020). Deficiencies in complex IV are primarily due to mutations in mtDNA and nDNA-encoded assembly factors, resulting in hypertrophic cardiomyopathy, ataxia, lactic acidemia, and Leigh syndrome (Antonicka et al, 2003; Diaz et al, 2011; Valnot et al, 2000).

Although there are supportive therapies to mitigate the symptoms of some mitochondrial diseases, there is currently no curative treatments available (Baertling et al, 2014; Hanaford et al, 2022). In recent years, there has been a heightened interest in the use of viral vectors to treat rare disorders, with several options currently on the market, and mitochondrial diseases are suitable candidates. We and others have tested AAV-vector-based gene replacement therapies in mouse models of OXPHOS defects (Bottani et al, 2020; Corrà et al, 2022; Ling et al, 2021; Pereira et al, 2020; Reynaud-Dulaurier et al, 2020; Silva-Pinheiro et al,

[1]Neuroscience Graduate Program, University of Miami Miller School of Medicine, Miami, USA. [2]Department of Neurology, University of Miami Miller School of Medicine, Miami, USA. [3]Department of Ophthalmology, University of Miami Miller School of Medicine, Miami, USA. [4]Department of Cell Biology, University of Miami Miller School of Medicine, Miami, USA. [5]Present address: Mitobridge Inc, Cambridge, MA, USA.✉E-mail: cmoraes@med.miami.edu

2020). AAV-PHPB-*hNdufs4* was shown to extend the life of a *Ndufs4* KO model (Corrà et al, 2022; Reynaud-Dulaurier et al, 2020; Silva-Pinheiro et al, 2020).

In the present study, we have utilized previously characterized neuronal-specific mouse models of CI and CIV deficiencies to investigate the effectiveness of AAV-PHP.eB as a delivery method for a gene replacement (Diaz et al, 2012; Fukui et al, 2007; Peralta et al, 2020). Retro-orbital injection of the AAV in juvenile mice essentially corrected their phenotypic behavior and extended lifespan in both Ndufs3- and COX10-conditional neuronal knock-out (nKO) mice. Unexpectedly, only a minority of neurons expressed the transgenes in the conditions applied.

# Results

## Retro-orbital injections of recombinant AAV-PHP.eB restored deleted OXPHOS genes, prolonged lifespan and maintained weight in OXPHOS nKO mice

We tested whether the restoration of a missing OXPHOS complex in neurons could protect mice against mitochondrial encephalopathy. Previously, our lab characterized neuron-specific mouse models of OXPHOS complex I and complex IV deficiency. These conditional neuronal knockouts (nKO) were created using CamKIIα-Cre and floxed *Ndufs3* and *Cox10* alleles, respectively. NADH-ubiquinone oxidoreductase iron-sulfur protein 3 (NDUFS3) is a nuclear encoded subunit of Complex I, critical to Complex I assembly and function (Procaccio et al, 2000; Vogel et al, 2007). COX10 is a nuclear encoded protein involved in heme *a* biosynthesis, and critical for the assembly of Complex IV (Mogi

et al, 1994). Heme *a* is an indispensable cofactor for the maturation of COX1, a catalytic subunit of Complex IV. The *Ndufs3*- and *Cox10*-nKO mice experience shortened lifespans, progressive weight and neuronal loss, and impaired motor coordination and balance (Diaz et al, 2012; Peralta et al, 2020).

To replace the missing genes in neurons, we used recombinant AAV-PHP.eB viruses. This AAV capsid subtype has been shown to effectively deliver genes to neurons of C57BL6 mice (Chan et al, 2017). For simplicity, henceforth we will refer to AAV-PHP.eB as "AAV". The recombinant genes contain the human synapsin promoter (hSYN) driving the neuronal expression of cDNA coding for *Ndufs3*, *Cox10*, or *eGFP*. Figure 1A illustrates the structure of the constructs used. The recombinant AAVs were systemically delivered to 18–21-day-old nKO mice via a single retro-orbital injection (i.v.), as previously described (Bacman et al, 2018; Pereira et al, 2020; Yardeni et al, 2011). We treated and analyzed five groups: *Ndufs3*-nKO mice receiving AAV-*eGFP* (*Ndufs3*-nKO+*eGFP*), *Ndufs3*-nKO mice receiving AAV-*Ndufs3* (*Ndufs3*-nKO+*Ndufs3*), *Cox10*-nKO mice receiving AAV-*eGFP (Cox10*-nKO+*eGFP*), *Cox10*-nKO mice receiving the AAV-*Cox10 (Cox10*-nKO+*Cox10*), and CamKIIα-Cre hemizygous mice (WT). Both males and females were treated.

The different groups were analyzed for body weight, rotarod, and open field behaviors weekly after the injections. The timeline of the experiments is depicted in Fig. 1B. As observed during the initial characterization, nKO mice injected with AAV-*eGFP* exhibited hunched posture and began to lose weight between 4 and 4.5 months of age (Fig. 2A,C,E). On the other hand, after administration of the respective conditionally deleted genes, both OXPHOS-nKO models showed normal posture and no weight loss (Fig. 2A,C,E).

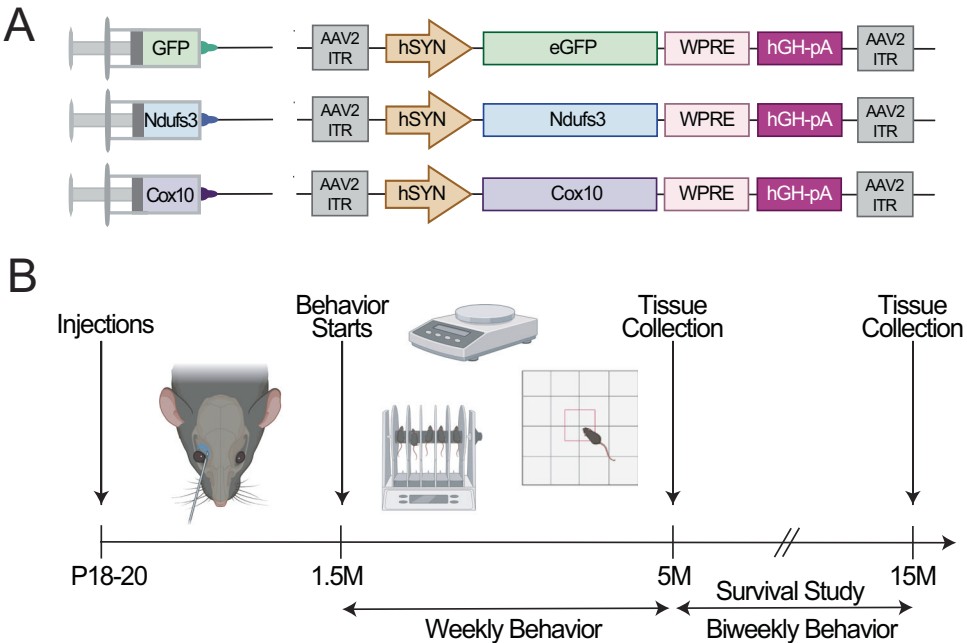

**Figure 1. Experimental design of OXPHOS gene replacement.**

(A) Design of AAV-PHP.eB viral construct. Viral constructs contain the human synapsin (hSYN) for neuronal specificity. Constructs contain either GFP (nKO control), *Ndufs3*, or *Cox10*. (B) Experimental timeline for AAV administration and analyses.

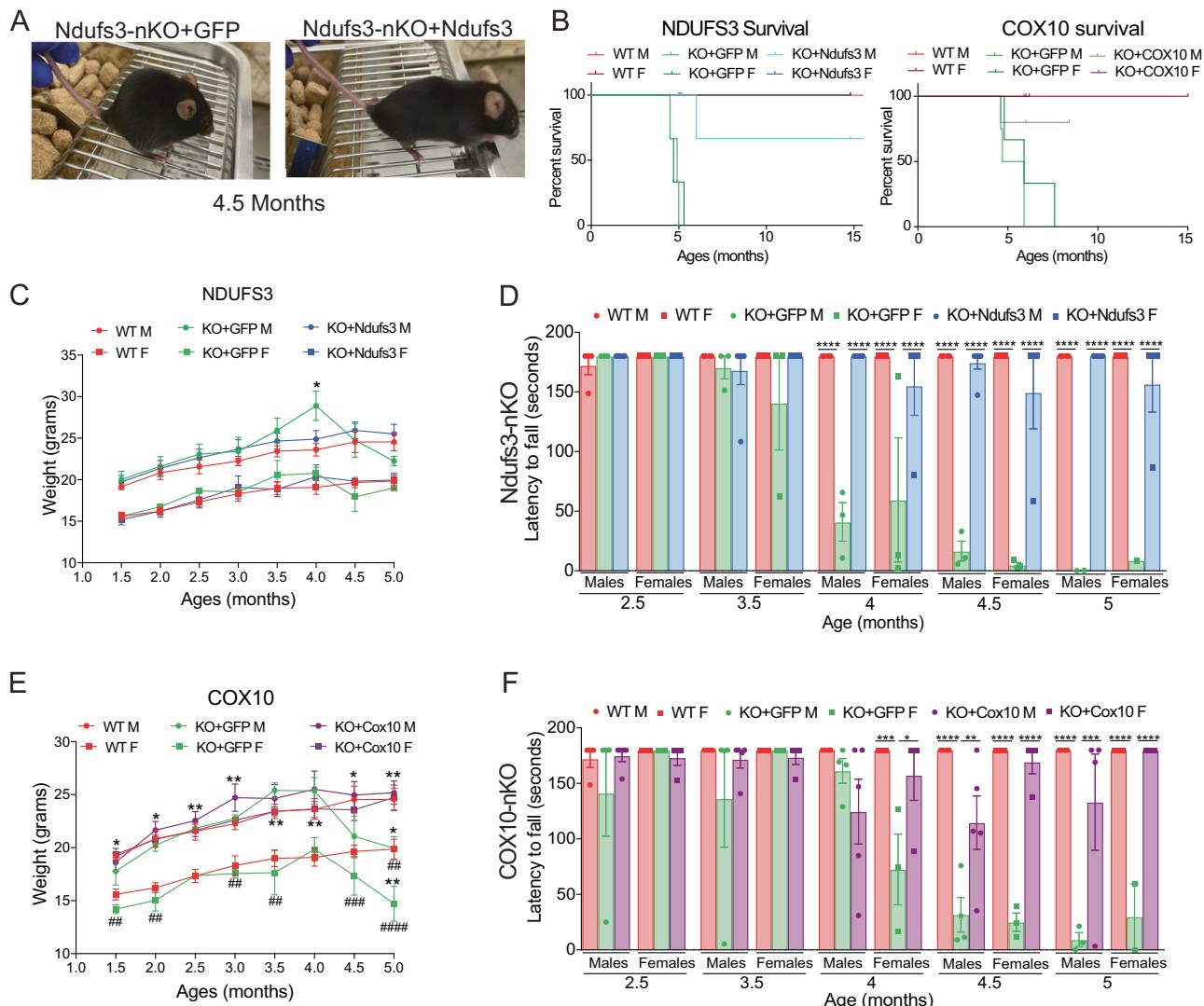

**Figure 2. OXPHOS gene replacement prevents behavioral changes.**

(A) Representative image of 4.5-month-old nKO mice injected with AAV-PHP.eB-hSYN-*eGFP* or AAV-PHP.eB-hSYN-*Ndufs3*, exhibiting hunched posture. (B) Survival curves of *Ndufs3*-nKO (left) and *Cox10*-nKO (right) mice. (C) Weekly weights of *Ndufs3*-nKO cohort over the course of the age-matched study. (D) Rotarod performed by *Ndufs3*-nKO cohort at 2.5, 3.5, 4, 4.5, and 5 months of age. (E) Weekly weights of *Cox10*-nKO cohort over the course of the age-matched study. (F) Rotarod performed by *Cox10*-nKO cohort at 2.5, 3.5, 4, 4.5, and 5 months of age. Data information: In (C–F), data are represented as mean ± SD. For (C, D), WT males (red circles, n = 4), WT females (red squares, n = 6), KO + GFP males (green circles, n = 3) KO + GFP females (green squares, n = 3), KO+Ndufs3 males (blue circles, n = 6), and KO+Ndufs3 females (blue squares, n = 4). For (E–F), WT males (red circles, n = 4), WT females (red squares, n = 6), KO + GFP males (green circles, n = 4) KO + GFP females (green squares, n = 3), KO+Cox10 males (purple circles, n = 5), and KO+Cox10 females (purple squares, n = 4). n values were calculated using two-way ANOVA, with Tukey's multiple comparisons test. In (C, E), * Compared to WT. # Compared to KO + OXPHOS. $P(*/#) = 0.0332$, $P(**/\#\#) = 0.0021$, $P(***/\#\#\#) = 0.0002$, $P(****/\#\#\#\#) < 0.0001$. Exact $P$ values are listed in Appendix Table S1. Source data are available online for this figure.

*Ndufs3*-nKO mice develop a severe encephalopathy and need to be sacrificed between 4.5 and 5 months of age (Peralta et al, 2020). Although WT and *Ndufs3*-nKO+*Ndufs3* appeared normal and healthy at this age, they were also sacrificed at 5 months of age for molecular and histological comparisons with the nKO model. We allowed some mice to age beyond the 5-month timepoint and observed that AAV-*Ndufs3* injections significantly extended

survival (Fig. 2B). Three of these mice were sacrificed at 15 months of age with no detectable encephalopathy phenotype. A fourth mouse died at 6 months of age for reasons apparently unrelated to neurological problems.

Although *Cox10*-nKO mice can live up to 8–10 months, quality of life progressively declines around 5 months of age (Diaz et al, 2012). Similarly to the *Ndufs3* model, mice were sacrificed and

analyzed at 6 months of age. A small cohort of mice was allowed to age beyond the 6-month timepoint, and we observed that AAV-*Cox10* injections greatly extended survival, to 15 months (Fig. 2B).

## AAV-mediated gene therapy improved motor function, balance, and coordination in OXPHOS-deficient mice

*Ndufs3*-nKO mice lose motor coordination and balance at 3 months of age (Peralta et al, 2020). In the current study, we analyzed the motor and coordination phenotypes through rotarod and open field tests. At 3.5 months of age, clear deficits were noticed in the *Ndufs3*-nKO and *Cox10*-nKO mice on the rotarod that became significant at 4 months of age and continued until mice were terminal. In contrast, *Ndufs3*-nKO+*Ndufs3* behaved like WT mice (Fig. 2D). *Cox10*-nKO+*Cox10* also had greatly improved motor balance (Fig. 2F).

The open field test is less sensitive to detect behavioral changes in these models because the OXPHOS nKO models undergo a period of hyperactivity that masks decrease in activity or vertical counts. Nonetheless, such impairment becomes evident by 4–4.5 months depending on the model (Appendix Figure S1). Improved performance was clear, as *Ndufs3*-nKO+*Ndufs3*, *Cox10*-nKO+*Cox10* and WT mice did not show differences in exploratory or rearing patterns over time (Appendix Figure S1).

## NDUFS3 levels in the CNS were increased in Ndufs3-nKO mice injected with AAV-PHP.eB-Ndufs3

To quantify changes in NDUFS3 protein in *Ndufs3*-nKO mice, we performed western blots of cortex and hippocampal homogenates of 5-month-old male mice. In cortices of 5-month-old *Ndufs3*-nKO+*eGFP* mice, NDUFS3 levels were depleted to ~40%, which were restored to ~90% with the AAV-*Ndufs3* treatment (Fig. 3A,B). We observed a similar increase of NDUFS3 levels in the hippocampus, from 17 to 92% (Fig. 3D,E). We also analyzed another subunit of Complex I, NDUFB8, in both the cortex and hippocampus. NDUFB8 is known to be unstable if Complex I is not assembled and thereby is a useful surrogate to Complex I holoenzyme (Lazarou et al, 2009; Peralta et al, 2020). NDUFB8 was depleted to approximately 34% and 48% in cortex and hippocampus, respectively, of *Ndufs3*-nKO mice. NDUFB8 levels were restored to 69% and 89% in these brain regions with gene therapy (Fig. 3B,E). In cortex and hippocampal homogenates of 5-month-old female mice, NDUFS3 levels increased from 32% to 60% and 36% to 42%, respectively (Fig. EV1A,B,E,F). A bigger change was observed with the marker NDUFB8 in both cortex (36% to 71%) and hippocampus (44% to 81%) (Fig. EV1A,B,E,F).

Western Blot analyses for proteins of complexes II, III, and IV in the *Ndufs3* nKO model did not show major changes in cortex and hippocampus (Fig. 3C,F). An increase of COX1 (CIV) in the hippocampus of *Ndufs3*-nKO mice was reverted by the gene replacement (Fig. 3F). In the cortex of female mice, we detected increases in SDHA (CII) and UQCRC1 (CIII) that were resolved with AAV treatment (Fig. EV1A,C). A similar trend was observed with hippocampal levels of SDHA, however, UQCRC1 levels were decreased in both nKO groups (Fig. EV1E,G). Examining another mitochondrial marker (VDAC), an outer mitochondrial membrane protein, did not suggest an increase in mitochondrial mass in cortex; however, there was an increase in male hippocampus

homogenates and a decrease in female hippocampal homogenates (Figs. 3G and EV1D,H). Finally, *Ndufs3*-nKO mice showed a trend towards an increase in mtDNA copy number (Appendix Figure S2).

To verify the reintroduction of NDUFS3 protein in cortical neurons, we performed immunofluorescence imaging with anti-NDUFS3 and NeuN. Approximately 50% of NeuN-positive cells in *Ndufs3*-nKO+*Ndufs3* cortex had clear, mitochondrial localization of NDUFS3 staining, compared to 10% in nKO+*eGFP* and 80% in WT mice (Fig. 3H). Therefore, after normalization to the levels observed in WT samples, ~34% NeuN-positive somas express NDUFS3 from the AAV-PHP.eB.

## COX1 levels are increased in the CNS of COX10-nKO mice injected with AAV-PHP.eB-Cox10

To quantify changes in COX10 protein in *Cox10*-nKO mice, we performed Western blot of cortex and hippocampal homogenates of 5-month-old male mice. As a surrogate for COX10 expression, we probed for COX1, as COX10 expression is essential for COX1 maturation and stability (Mogi et al, 1994) (Fig. 4A,E). Although we typically normalize to total protein loading, we observed a significant increase of VDAC in the cortex of *Cox10*-nKO+*eGFP* mice (Fig. 4D). This would suggest an increase in mitochondrial mass which would be reflected in the levels of OXPHOS subunits; therefore, we compared Complex IV subunits to VDAC after normalization for total protein loading. In the cortex, COX1 and COX4 (a nuclear-encoded Complex IV subunit) levels are clearly depleted in *Cox10*-nKO+*eGFP* mice and restored in treated mice (Fig. 4A,B). In the hippocampus of *Cox10*-nKO+*eGFP* animals, COX1 was comparable to WT levels, while COX4 was depleted (Fig. 4E,F). Interestingly, VDAC was not increased in hippocampus samples, whereas mtDNA trended higher, but it was not significant (Fig. 4H,I). Hippocampal homogenates of *Cox10*-nKO+*Cox10* mice show comparable levels of COX1 without the accompanying increase in mtDNA levels (Fig. 4E,F).

Western Blot analyses for subunits of complexes I, II and III showed a mild increase in SDHA (CII) and UQCRC1 (CIII) in the cortex and hippocampus of nKO+*eGFP* animals, which was partially resolved in *Cox10*-nKO+*Cox10* animals, as SDHA still appeared to be elevated in the hippocampus (Fig. 4C,G).

To verify reintroduction of COX10 protein in cortical neurons, we performed immunofluorescence imaging with anti-COX1 and NeuN. Approximately 60% of NeuN-positive cells in *Cox10*-nKO+*Cox10* cortex had clear, mitochondrial localization of COX1 staining, compared to 30% nKO+*eGFP* and 70% in WT mice (Fig. 4J).

## Reintroduction of deleted OXPHOS proteins leads to proper assembly and function of affected complexes

Our group had previously observed that lack of NDUFS3 significantly impaired Complex I assembly and supercomplex formation. To analyze Complex I levels, we analyzed cortical homogenates on BN-PAGE and probed for NDUFA9, a subunit of Complex I. The *Ndufs3*-nKO+*eGFP* had a marked decrease in Complex I levels in the supercomplex CI + CIII and on its own (~49% compared to controls). With AAV-*Ndufs3* treatment, CI levels were restored to 93% in males, and 77% in females (Figs. 5A,B and EV1I,J). Probing for UQCRC1, a

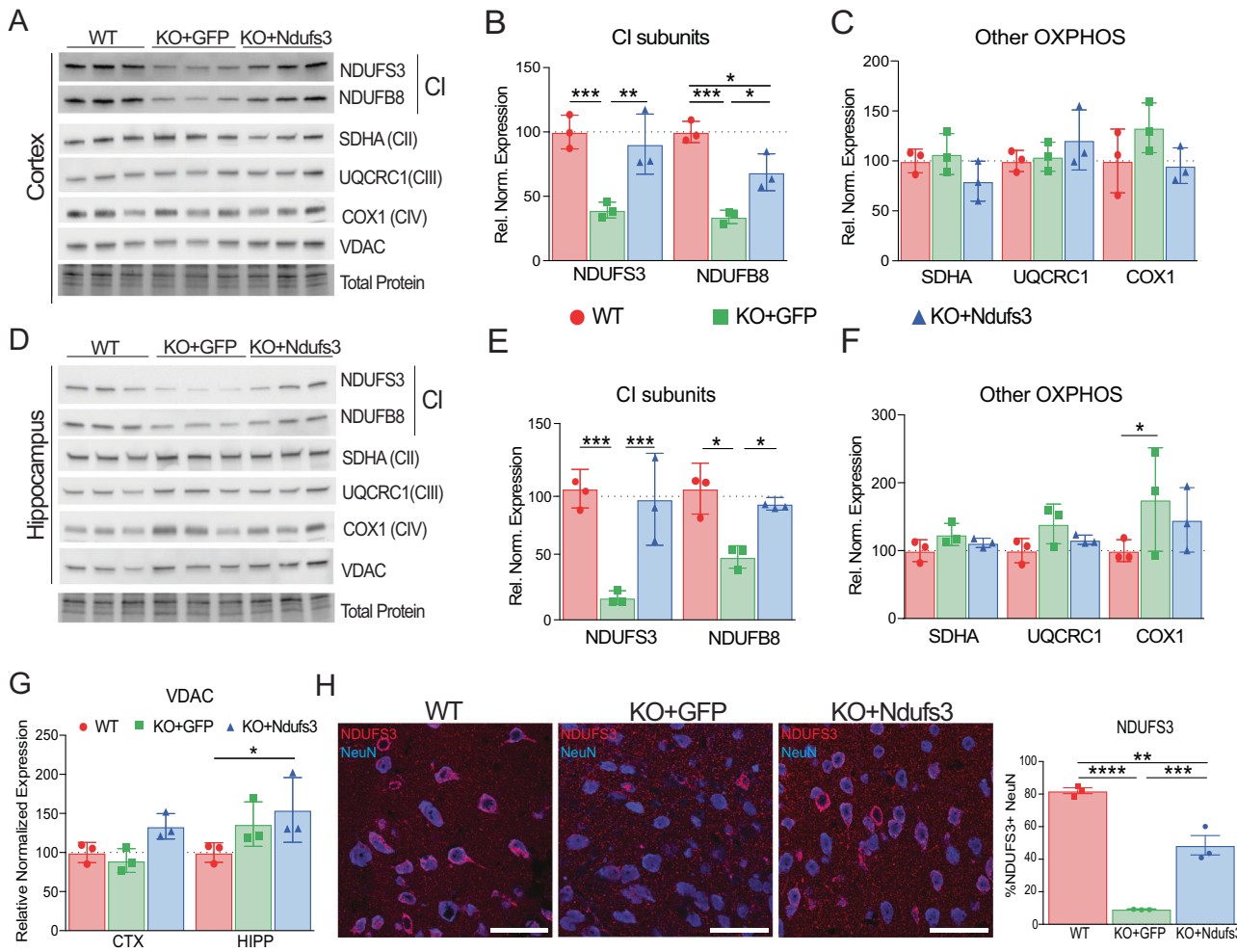

**Figure 3. Restoration of NDUFS3 in *Ndufs3*-nKO mice.**

(A–G) Western blots and relative quantifications of protein homogenates from cortex and hippocampus of 5-month-old wild-type (WT), *Ndufs3*-nKO+eGFP, and *Ndufs3*-nKO+*Ndufs3* male mice probed for NDUFS3 and NDUFB8 (Complex I subunits), SDHA (Complex II subunit), UQCRC1 (Complex III subunit), COX1 (Complex IV subunit), and VDAC (mitochondrial membrane protein). Total protein loading was used as loading control. All protein loading staining and their respective blots are shown in Appendix Fig. S5. (H) Immunohistochemical images and quantification of NDUFS3 and NeuN in cortex of 5-month-old male mice. Scale bar is 50 μm. Data information: In (B, C, E–G), data are represented as mean ± SD (n = 3/group). P values were calculated using two-way ANOVA, Tukey's multiple comparisons test. In (H), data are represented as mean ± SD (n = 3/group). P values were determined by one-way ANOVA, Tukey's multiple comparisons test. P(*) = 0.0332, P(**) = 0.0021, P(***) = 0.0002, P(****) < 0.0001. Exact p-values can be found in Appendix Table S1. Source data are available online for this figure.

subunit of Complex III, the disruption of the CI + CIII supercomplex is also clear in *Ndufs3*-nKO+*eGFP* mice (Figs. 5A,B and EV1I,J). In addition to recovery of Complex III levels, Complex IV was increased in *Ndufs3*-nKO+*Ndufs3* mice (Fig. 5B, right). To further confirm functionality of assembled Complex I, we performed an in-gel activity (Fig. 5C). We observed a marked decrease in functional Complex I in the nKO+*eGFP* (48% ± 10.97) that was rescued by the gene therapy (91.47% ± 9.56) (Fig. 5D, left). Similar results were observed with female animals (47% to 77%) (Fig. EV1K,L). There were no differences in Complex IV activity between the groups, in either sex (Figs. 5D and EV1L).

COX10 depletion led to a progressive Complex IV defect, significantly impairing Complex IV enzymatic activity (Fig. 5E,F). Interestingly, in addition to a reduction in Complex IV levels in *Cox10*-nKO+*eGFP* mice, we observed a decrease in the CI + CIII supercomplex when probing for NDUFA9 and an increase in "free"

Complex III when probing for UQCRC1 (Fig. 5E,F). Therefore, Complex IV deficiency led to an instability of supercomplexes (Novack et al, 2020). Complex IV levels were restored from 59% to 127% with AAV-PHP.eB-*Cox10* treatment (Fig. 5F, left). Complex I (Fig. 5F, middle) and Complex III (Fig. 5F, right) levels were also restored to WT levels in *Cox10*-nKO+*Cox10* mice. Functionality of Complex IV was confirmed in the rescued brains, from 71.15% ± 11.78 to 110.6% ± 21.4 (Fig. 5G,H).

## AAV-PHP.eB-mediated OXPHOS restoration reduces neuroinflammation and neuronal loss in cortex and hippocampus of treated OXPHOS nKO mice

In previous studies, we showed that deletion of *Ndufs3* was associated with neuronal death in the hippocampus at 4 months, despite no major differences in brain weight (Diaz et al, 2012;

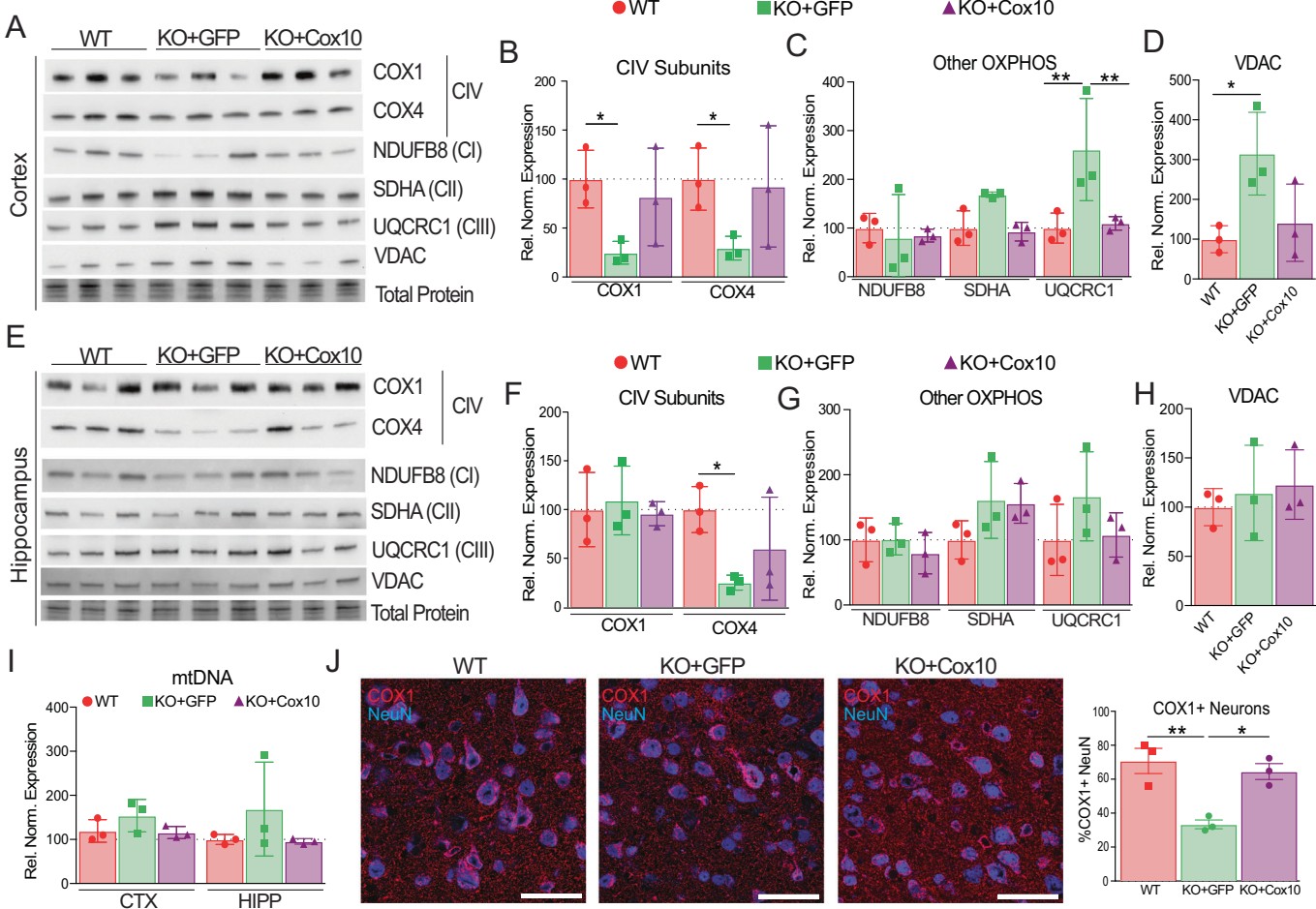

**Figure 4. Restoration of COX10 in *Cox10*-nKO mice.**

(A–H) Western blots and relative quantifications of protein homogenates from cortex and hippocampus of 5-month-old wild-type, *Cox10*-nKO+*eGFP*, and *Cox10*-nKO+*Cox10* male mice probed for COX1 and COX4 (Complex IV subunits), NDUFB8 (Complex I subunit), SDHA (Complex II subunit), UQCRC1 (Complex III subunit), and VDAC (mitochondrial membrane protein). Total protein loading was used as loading control. All protein loading staining and their respective blots are shown in Appendix Fig. S5. (I) mtDNA levels measured by digital PCR in DNA extracted from cortex and hippocampus of 5-month-old mice. (J) Immunohistochemical images and quantification of COX10 and NeuN in cortex of 6-month-old male mice. Scale bar is 50 μm. Data information: In (B, C, F, G, I), data are represented as mean ± SD ($n = 3$/group). P values were calculated using two-way ANOVA, Tukey's multiple comparisons test. In (D, H, J), data are represented as mean ± SD ($n = 3$/group). P values were determined by One-way ANOVA, Tukey's multiple comparisons test. $P(*) = 0.0332$, $P(**) = 0.0021$. Exact P values are listed in Appendix Table S1. Source data are available online for this figure.

Peralta et al, 2020). Additionally, *Ndufs3*-nKO mice displayed marked glial activation in the cortex and hippocampus. In the current study, we performed immunofluorescence imaging with antibodies against glial acidic fibrillary protein (GFAP, activated astrocyte marker) and IBA1 (microglia marker) on cortical slices of 5-month-old animals. GFAP and IBA1 staining was decreased in *Ndufs3*-nKO+*Ndufs3* slices compared to nKO+*eGFP* (Fig. 6A,B). Western blot analysis supported these data, with *Ndufs3*-nKO+*Ndufs3* mice displaying significantly decreased GFAP and IBA1 levels compared to nKO+*eGFP* (Figs. 6C,D and EV2A,B). To analyze neuronal loss, we measured TUJI levels by western blot and found slight decreases in both cortex and hippocampus that were mitigated in *Ndufs3*-nKO+*Ndufs3* mice (Fig. 6C,D). We did not detect any changes in TUJI levels in *Ndufs3*-nKO female mice (EV2A,B). As previously reported, no changes were observed in the brains weight of all three *Ndufs3* groups at 5 months (Fig. 6E).

In contrast to *Ndufs3*-nKO mice, *Cox10*-nKO mice exhibit a significant decrease in brain mass with obvious cortical atrophy at 6 months of age (Diaz et al, 2012). Treatment with AAV-PHP.eb-*Cox10* prevented cortical mass loss (Fig. 7A,B). *Cox10*-nKO mice also experience dramatic glial activation in the cortex and hippocampus, as seen via immunofluorescent imaging, and supported by western blot quantifications (Fig. 7C–F). Although mice receiving AAV-PHP.eB-*Cox10* displayed a decrease in glial markers in cortex homogenates, GFAP and IBA1 remain elevated in the hippocampus (Fig. 7F). Despite notable differences in appearance and mass of the cortex, *Cox10*-nKO+*eGFP* mice did not display drastic changes in TUJI levels on western blot analysis; however hippocampal homogenates displayed a mild decrease in both *Cox10*-nKO groups (Fig. 7E,F). Via NeuN immunofluorescent staining, we observed hippocampal degeneration, but no apparent differences in staining throughout the cortex (Fig. 7G).

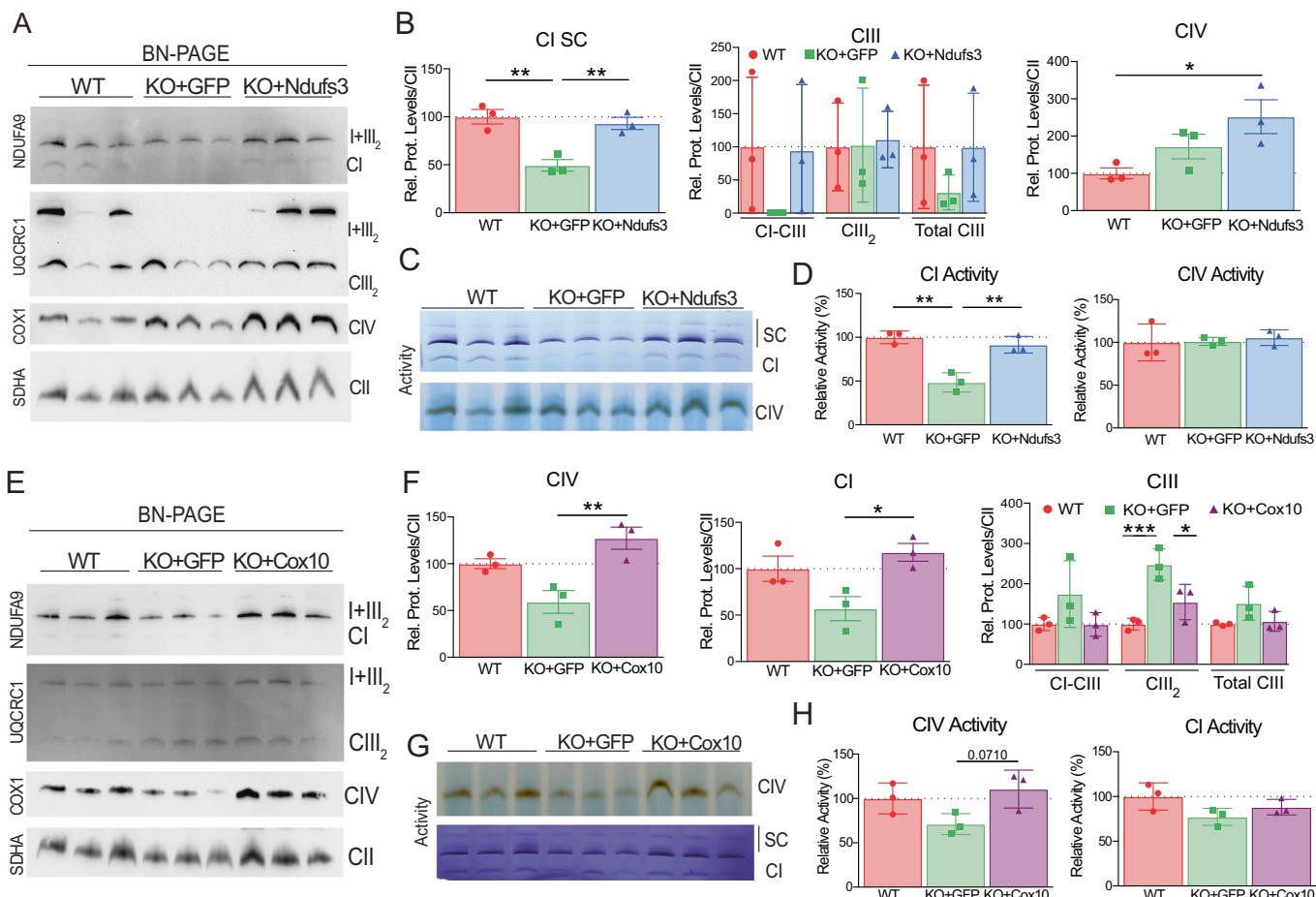

**Figure 5. Complex assembly and function in nKO mice is restored by CNS gene therapy.**

(A, B) BN-PAGE and relative quantifications of steady-state levels of respiratory complexes, normalized to CII levels, in cortical homogenates of *Ndufs3*-nKO mice. (C, D) BN-PAGE in gel activity and relative quantifications of enzymatic activity in cortical homogenates of *Ndufs3*-nKO mice. (E, F) BN-PAGE and relative quantifications of steady-state levels of respiratory complexes, normalized to CII levels, in cortical homogenates of *Cox10*-nKO mice. (G, H) BN-PAGE in gel activity and relative quantifications of enzymatic activity in cortical homogenates of *Cox10*-nKO mice. Data information: In (B, D, F, H), data are represented as mean ± SD ($n = 3$/group). $P$ values were calculated using One-way ANOVA, with the exception of CIII panels, which were calculated using two-way ANOVA, both with Tukey's multiple comparisons test. $P(*) = 0.0332$, $P(**) = 0.0021$, $P(***) = 0.0002$. Exact $P$ values are listed in Appendix Table S1. Source data are available online for this figure.

## Early therapeutic intervention improves quality of life and extends survival to at least 15 months of age in nKO mice

As previously mentioned, we followed a small cohort of nKO mice and their WT littermates to 15 months of age. These mice maintained a healthy weight and posture (Fig. EV3A). On the rotarod, both nKO+OXPHOS mice performed in a manner comparable to WT controls, excluding one *Cox10*-nKO+*Cox10* mouse who had some difficulties due to increased weight (Fig. EV3B). In the open field analysis, *Ndufs3*-nKO+*Ndufs3* mice tended to have more exploratory behavior compared to both WT and *Cox10*-nKO+*Cox10* mice, as noted by vertical counts (Fig. EV3C). Following euthanasia and dissection, we found no significant differences between the groups in terms of brain weight or appearance.

To quantify levels of the restored protein, we performed western blots for NDUFS3 and COX1 in both cortex and hippocampal homogenates. In both models, the targeted protein was comparable to WT levels, suggesting effective long-term expression of the AAV (Fig. EV3D,E). To confirm sustained stability and functionality of the targeted Complexes, we performed BN-PAGE and in-gel activity assays for Complexes I and IV. Complex I and IV levels and enzymatic activity were comparable to WT levels (Fig. EV3F–I).

Immunofluorescent staining for neuroinflammatory markers GFAP and IBA1 also did not show major differences between treated and WT animals (Fig. EV4A,B). Western blots showed that *Ndufs3*-nKO+*Ndufs3* animals do have a mild, though not significant increase in GFAP levels compared to their WT counterparts (Fig. EV4C,D). Additionally, *Cox10*-nKO+*Cox10* animals had an increase in hippocampal IBA1 (Fig. EV4C,D). No significant differences were observed concerning neuronal content or cortical mass (Fig. EV4C–E). This data confirms that the neuroinflammation seen at 5–6 months of age is resolved and maintained in older ages of treated mice.

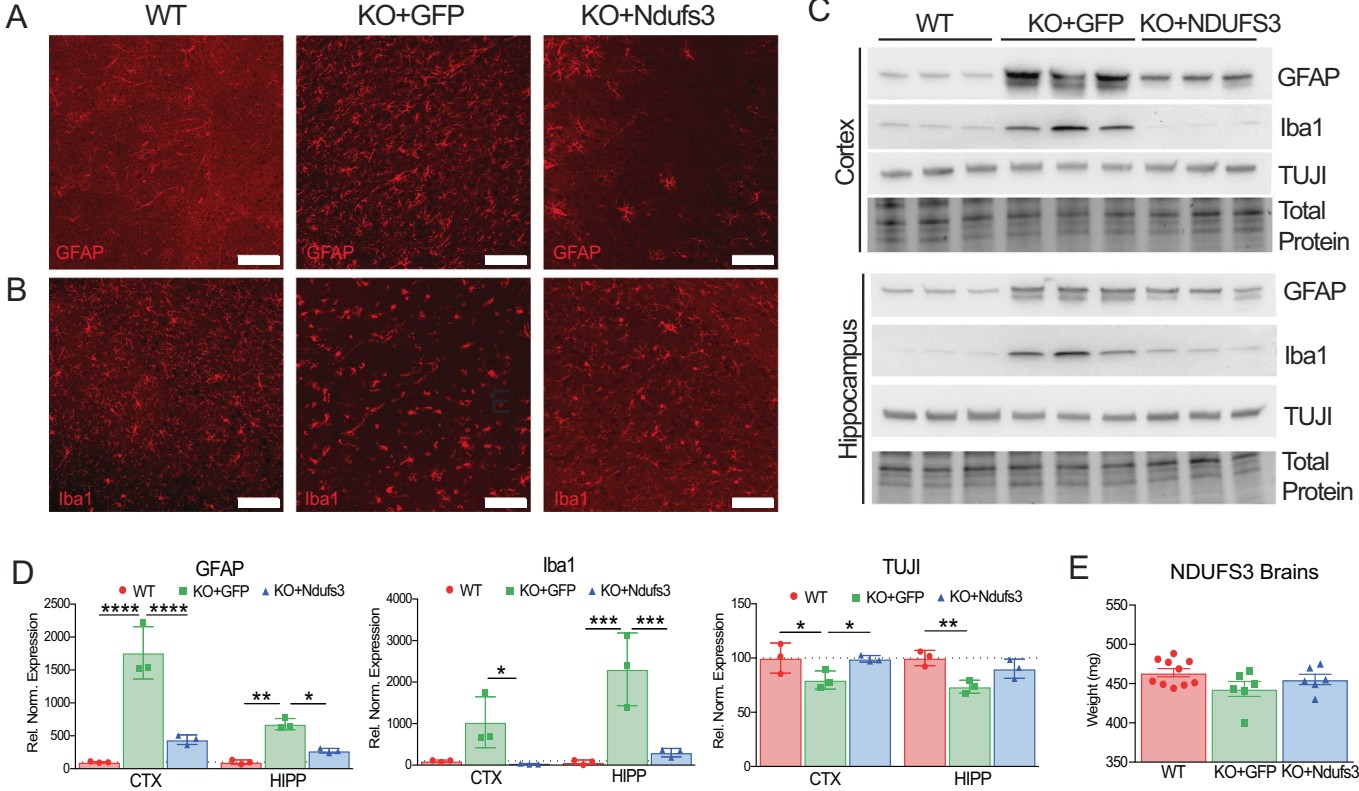

**Figure 6. Neuroinflammation in *Ndufs3*-nKO mice is improved by CNS gene therapy.**

(A) Immunohistochemical images of GFAP in motor cortex of 5-month-old male mice. Scale bar is 100 μm. (B) Immunohistochemical images of IBA1 in motor cortex of 5-month-old male mice. Scale bar is 100 μm. (C, D) Western blots and relative quantification of protein homogenates from cortex and hippocampus of males WT, KO+eGFP, and KO+*Ndufs3* mice at 5 months of age, probing for astrocyte activation (GFAP), microglial marker IBA1, and neuronal marker TUJ1. Total protein loading was used as loading control. All protein loading staining and their respective blots are shown in Appendix Fig. S5. (E) Brain weight of 5-month-old wild-type, *Ndufs3*-nKO+*eGFP*, and *Ndufs3*-nKO+*Ndufs3* mice, both male and female. Data information: In (D), data are represented as mean ± SD (n = 3/group). P values were calculated using two-way ANOVA, with Tukey's multiple comparisons test. In (E), data are represented as mean ± SD. WT (males n = 7, females n = 4), KO + GFP (males n = 3, females n = 3), and KO+Ndufs3 (males n = 3, females n = 3). P values were determined by one-way ANOVA, with Tukey's multiple comparisons test. P(*) = 0.0332, P(**) = 0.0021, P(***) = 0.0002, P(****) < 0.0001. Exact P values are listed in Appendix Table S1. Source data are available online for this figure.

## Expression from AAV-PHP.eB-hSYN is limited to a subset of neurons but promotes robust CNS phenotypic correction

Our replacement gene constructs (*Ndufs3 and Cox10*) do not contain a tag, as it could affect folding, complex assembly or enzymatic function, therefore we used the AAV-*eGFP* virus as a surrogate to determine the extent of viral mediated expression of AAV-PHP.eB. We performed western blot analysis, probing for GFP, on homogenates of brain cortex, heart, liver, and kidney. As expected, we observed that GFP expression was only observed in the cortex sample in nKO+*eGFP* animals (Fig. 8A,B). We also quantified the number of viral particles delivered to cortex and hippocampus. This was performed by digital PCR with a custom primer/probe targeting the hSYN promoter, which is present only in the AAV, but not in the mouse genome. As expected, non-injected homogenates did not have hSYN DNA in either OXPHOS nKO model. Although similar AAV titers were injected, mice injected with AAV-*Ndufs3* had a higher AAV copy number compared to mice injected with AAV-*eGFP* (Fig. 8C, left), whereas this trended in the opposite direction for the *Cox10*-nKO mice

(Fig. 8C, right). These variabilities are not uncommon and reflect the percentage of viable viral particles in different preps. Because AAV copy number does not directly correlate to the number of infected cells, we performed immunohistochemistry on AAV-*eGFP*-injected animals using anti-GFP and anti-NeuN (Fig. 8D). We determined the percentage of GFP-/NeuN-double positive cells in the motor and piriform cortices and the hippocampus. Approximately 23% and 28% of NeuN-positive cells expressed GFP in *Ndufs3*-nKO and *Cox10*-nKO mice, respectively (Fig. 8E).

To better understand the extent of recovery by AAV-PHP.eB, we determined the levels of gene ablation over time, starting shortly after time of injection, using a three primer PCR to detect floxed, deleted and WT alleles. At 1-month of age, the percentage of *Ndufs3* recombination in the cortex and hippocampus homogenates was 47% and 59%, respectively. Recombination peaked at 3 months of age at about 60% in both cortex and hippocampus homogenates (Appendix Fig. S3A,B). Although Cre recombination plateau, we observed a steady decrease in NDUFS3 protein levels to approximately one-third of WT levels, as estimated by the ratio of NDUFS3 to SDHA (Appendix Fig. S3C,D). Together this implies that at least 50% of the cells in these regions have undergone

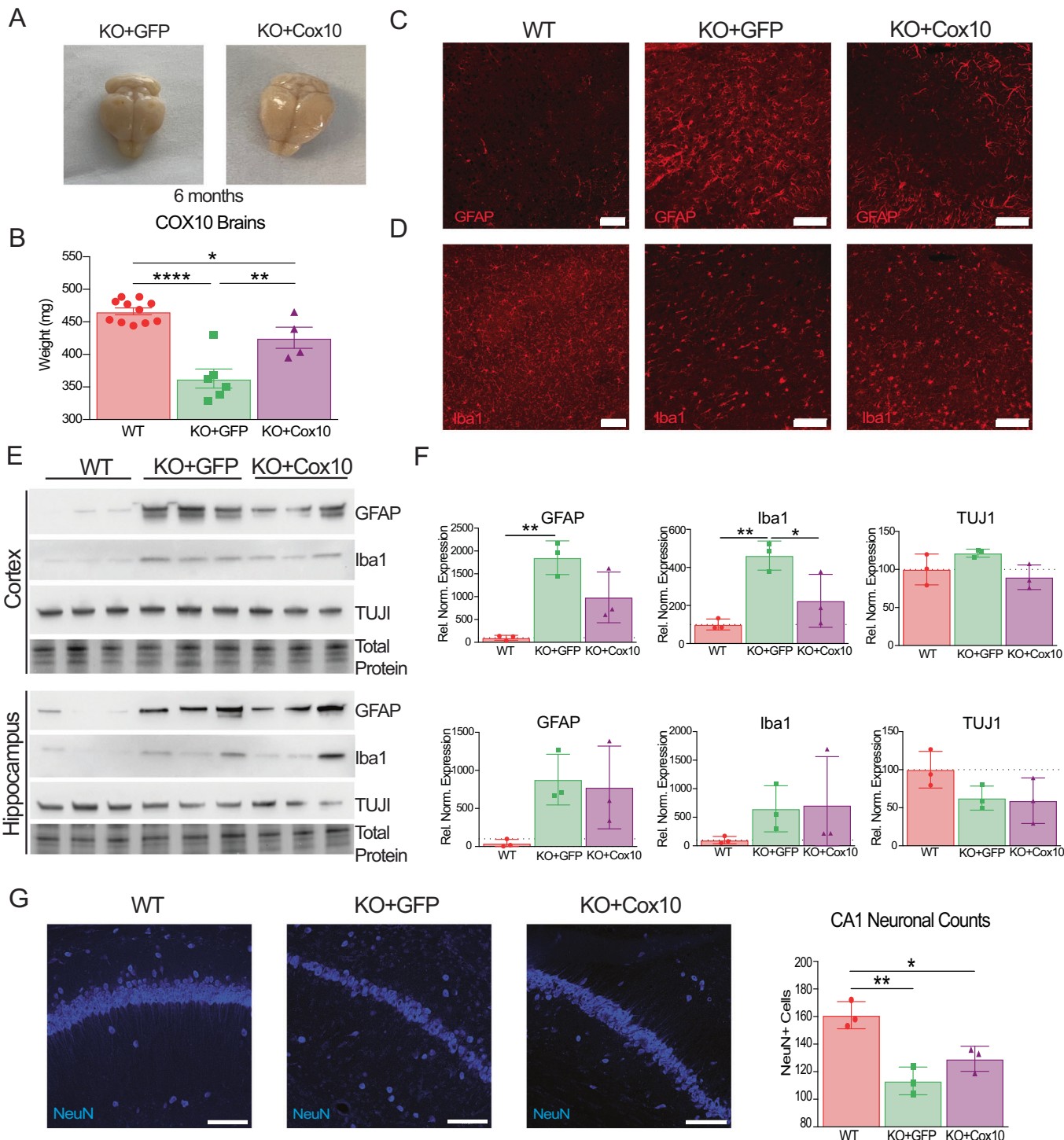

**Figure 7. Neuroinflammation in *Cox10*-nKO mice is improved by CNS gene therapy.**

(**A**) Gross brain morphology of 6-month-old *Cox10*-nKO+GFP and *Cox10*-nKO+*Cox10* mice. (**B**) Brain weight of 6-month-old wild-type, *Cox10*-nKO+GFP, and *Cox10*-nKO+*Cox10* mice. (**C**) Immunohistochemical images of GFAP in motor cortex of 6-month-old male mice. Scale bar is 100 μm. (**D**) Immunohistochemical images of Iba1 in motor cortex of 6-month-old male mice. Scale bar is 100 μm. (**E, F**) Western blots and relative quantification of protein homogenates from cortex and hippocampus of males WT, KO + GFP, and KO+Cox10 mice at 6 months of age, probing for GFAP, IBA1, and TUJ1. Total protein loading was used as loading control. All protein loading staining and their respective blots are shown in Appendix Fig. S5. (**G**) Immunohistochemical images and quantification of NeuN in CA1 region of hippocampus of 6-month-old male mice. Scale bar is 100 μm. Data information: In (**B**), data are represented as mean ± SD. WT (males $n = 7$, females $n = 4$), KO + GFP (males $n = 3$, females $n = 3$), and KO+Cox10 (males $n = 3$, females $n = 1$). In (**F, G**), data are represented as mean ± SD ($n = 3$/group). *P* values were calculated using one-way ANOVA, with Tukey's multiple comparisons test. $P(^*) = 0.0332$, $P(^{**}) = 0.0021$, $P(^{****}) < 0.0001$. Exact *P* values are listed in Appendix Table S1. Source data are available online for this figure.

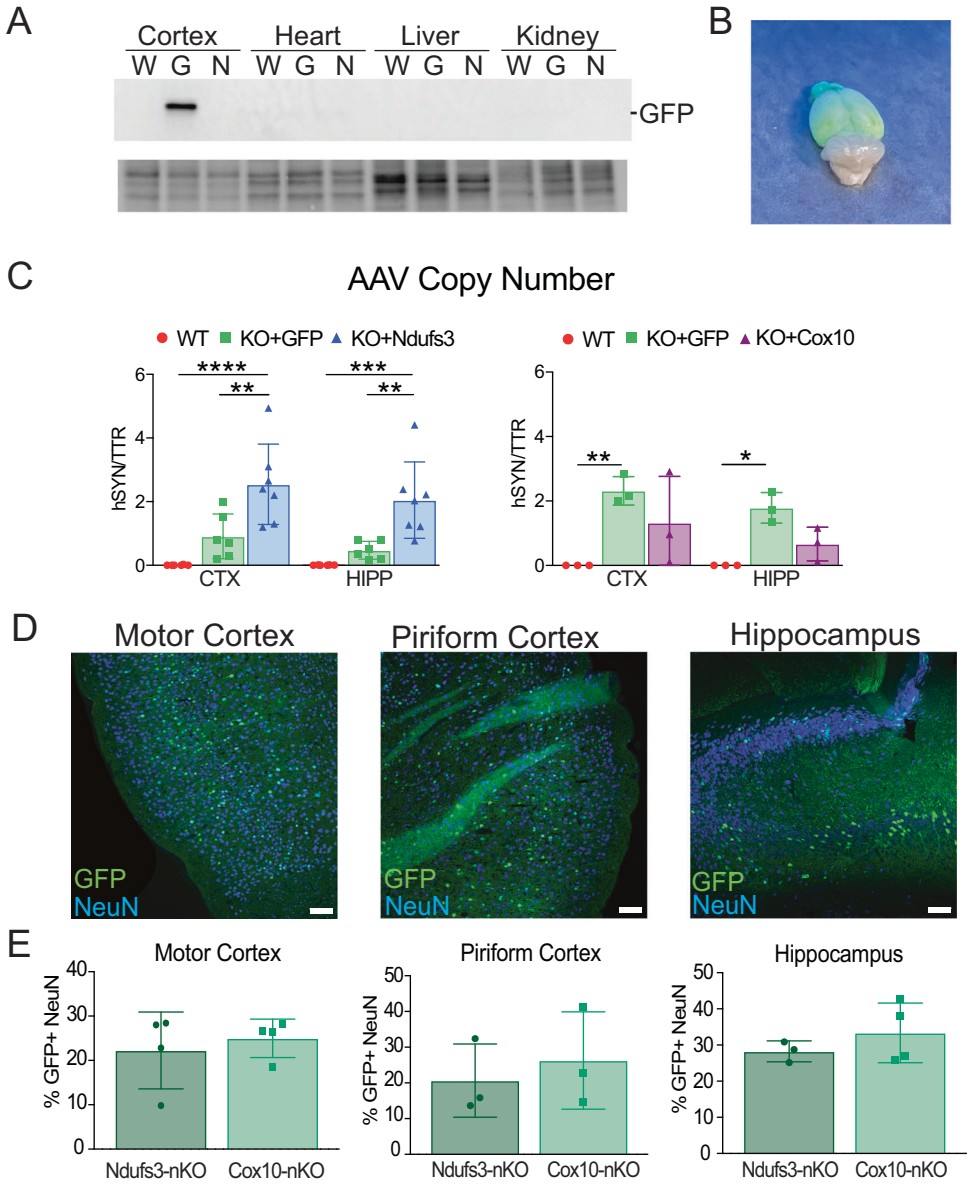

**Figure 8. Transgene expression of AAV-PHP.eB-hSYN viruses.**

(A) Western Blot of cortex, heart, liver, and kidney homogenates from wild-type (W), GFP-injected (G), and NDUFS3 (N) animals, probed for GFP. (B) AAV copy number as calculated by digital PCR analysis of the hSYN promoter. (C) Representative images of GFP-injected brain under black light, 6 months of age. (D) Representative frame containing GFP, NDUFS3, and NeuN staining. Scale bar is 50 μm. (E, F) Representative images and quantifications of GFP (green) expression in various brain regions of GFP-injected animals at 5–6 months of age. Scale bar is 100 μm. Data information: In (B), data are represented as mean ± SD. For *Ndufs3*-nKO, WT (males $n = 3$, females $n = 3$), KO + GFP (males $n = 3$, females $n = 3$), and KO+Ndufs3 (males $n = 4$, females $n = 3$). For *Cox10*-nKO, all samples are male, $n = 3$/group. P values were calculated using two-way ANOVA, with Tukey's multiple comparisons test. In (F), data are represented as mean ± SD ($n = 3$/group). P values were calculated using one-way ANOVA, with Tukey's multiple comparisons test. $P(*) = 0.0332$, $P(**) = 0.0021$, $P(***) = 0.0002$, $P(****) < 0.0001$. Exact P values are listed in Appendix Table S1. Source data are available online for this figure.

complete Cre recombination. Because homogenates contain a mixture of neuronal and glial subtypes, we can assume that most CamKIIα-positive neurons have both alleles deleted, leading to the severe Complex I or IV deficiency.

We next did a dose-response study of AAV.PHP.eB-hSYN-eGFP in 3-month-old mice. We retro-orbitally injected either a low ($1.5 \times 10^{11}$ vg), medium ($1.5 \times 10^{12}$ vg), or high ($7.5 \times 10^{12}$ vg) dose and collected the brain 8 weeks post injection. Our data showed

that the low dose, which was most similar to the doses used in the rescued mice, led to an average of 13% GFP+ NeuN+ while medium and high doses had comparable GFP+/NeuN+ expression with ~30% of NeuN+ cells expressing GFP (Appendix Figure S4A,B). There was a similar number of double positive cells even though AAV copy number in cortex was increased between the medium- and the high-titer dose (Appendix Figure S4C).

## Ndufs3 gene deletion in more than half of CamKIIα+ neurons does not cause detectable CNS phenotypes

Our gene replacement results suggested that a subset of OXPHOS-positive neurons protects the CNS from developing an encephalopathy. We further explored this concept by producing OXPHOS conditional nKO with an alternative, less active, CamKIIα-Cre strain. We found that the Cre-mediated flox ablation by B6.Cg-Tg(Camk2a-cre)T29-1Stl/J mice (known as T29-1) was less robust than the one by B6.Cg-Tg(Camk2a-cre)3Szi/J mice, the strain used in all the experiments described above. We followed the phenotype and floxed allele deletion of *Ndufs3*-nKO produced with the weaker T29-1 CamKIIα-cre driver. Even though the levels of *Ndufs3* gene ablation reached 20 and 35%, in cortex and hippocampus homogenates, respectively, the *Ndufs3*-nKO mice showed no behavioral phenotypes up to 6 months of age (Fig. EV5A–E). Western blot analysis of this new model showed 25 and 40% reduction in NDUFS3 in cortex and hippocampus homogenates, respectively (Fig. EV5F,G,J–K). Considering that these homogenates are not exclusively CamKIIα-positive neurons (i.e., they contain glia and other types of neurons), it is safe to assume that the *Ndufs3* deletion was substantially higher in neurons than the assays revealed. Therefore, the data with the weaker T29-1 CamKIIα-cre independently corroborate our AAV gene therapy rescue findings, showing that even a minority of OXPHOS-positive neurons prevents CNS failure, even when a substantial number of neurons are OXPHOS deficient.

## Discussion

We previously established two neuronal-specific knockouts of oxidative phosphorylation subunits that recapitulate features observed in patients with mitochondrial encephalopathies, including impaired locomotive function, balance, and coordination and a progressive neurodegeneration. Having previous success with gene replacement therapy in a skeletal muscle model (Pereira et al, 2020), we applied a modified approach to the neuronal models. We delivered a single injection of AAV-PHP.eB-hSYN virus retro-orbitally in pre-symptomatic, juvenile mice. Our results showed an essentially complete prevention of major phenotypes and restoration of NDUFS3 and COX10 levels in the nKO mice injected with AAV-PHP.eB-hSYN-*Ndufs3* and -*Cox10*, respectively. These models showed increased levels of complex I or complex IV, and a normalization of most of the phenotypes observed in untreated mice. The only exception was a small level of inflammation, which remained in the treated mouse brains.

Gene therapy has become a possible approach for treating rare disorders, and patients with mitochondrial diseases are promising candidates for this therapy (Ling et al, 2021). Notable preclinical work has been done in the field, as well, in models of Leigh Syndrome (Corrà et al, 2022; Ling et al, 2021; Pereira et al, 2020; Reynaud-Dulaurier et al, 2020; Silva-Pinheiro et al, 2020), Friedreich Ataxia (Gérard et al, 2014; Perdomini et al, 2014; Piguet et al, 2018), and mtDNA depletion (Bottani et al, 2014; Lopez-Gomez et al, 2021; Torres-Torronteras et al, 2018; Vila-Julià et al, 2020).

Genetic therapies for mitochondrial encephalopathies have been attempted in few models. AAV9 was used to treat both *Surf1* and *Ndufs4* KO models. In the former, AAV9 was injected intrathecally. This regimen resulted in a partially rescued complex IV activity in liver, brain, and muscle (Ling et al, 2021). Similarly, other studies have found that the phenotypic consequences of the *Ndufs4* KO model were corrected by AAV-PHP.B-mediated gene replacement. They found a delayed onset of neurodegeneration, and prolongation of the lifespan up to 1 year of age (Silva-Pinheiro et al, 2020). The same group showed that self-complementary AAV9, double injected (i.v. and intraventricular), showed even longer life extension (Corrà et al, 2022). However, the fraction of corrected neurons was not determined.

In the current study we employed AAV-PHP.eB, which is highly efficient for brain delivery in C57BL6 mice; however, when applied to non-human primates, and even other mouse strains, transduction efficiency is significantly lower (Goertsen et al, 2022). Therefore, the development of efficient and targeted viral variants is still needed for human CNS application. Engineering through CREATE (Cre recombination-based AAV targeted evolution), has identified additional CNS efficient AAV variants for non-human primates, such as CAP-B10 (Goertsen et al, 2022). In addition to vector optimization, dosage, promoter specificity, and delivery method must be considered. High dosages of viral therapy might increase the percentage of infected cells but may also induce toxicity or a strong immune response (Belbellaa et al, 2020). Altering the promoter can mitigate this, but because mitochondrial disease tends to have multi-systemic effects, a virus with widespread transduction and a ubiquitous promoter could be desirable. In the case of our study, we are aware the AAV-PHP.eB does transduce other cell types, however, the synapsin promoter has been shown to have strong neuronal specificity, which was ideal for robust expression in our neuronal knockout model (Kugler et al, 2003; Jackson et al, 2016).

Although our transgene was indistinguishable from the endogenous gene, making it impossible to determine the number of neurons expressing it from the transgene, the difference in NDUFS3 levels between cortex homogenates of *Ndufs3*-nKO and *Ndufs3*-nKO treated with AAV-PHP.eB-*Ndufs3* was 39% (average of males and females). Similarly, the difference in NDUFB8 in cortical homogenates was 35%. Estimates from NDUFS3-positive NeuN-positive neurons showed a 34% increase in NDUFS3 positivity in neurons by the treatment. We also used AAV-PHP.eB-*eGFP* as a surrogate to investigate the number of neurons infected by AAV-PHP.eB. This analysis showed that delivery of the transgene to be ~30% of neurons (GFP+/NeuN+).

As previously reported, neurodegeneration and gliosis was observed in *Ndufs3*-nKO mice once CI activity had dropped to ~25% of WT levels (Peralta et al, 2020). From these results, we can deduce that deletion levels and neuroinflammation are closely correlated in these models. Restoration of the deleted protein not only improves OXPHOS function, but also decreases neuroinflammation. Indeed, based on the number listed above, infection/expression of the missing gene in 30% of neurons increased Complex activity in both models by ~40% and triggers a significant reduction in gliosis.

Taken together, this study suggests that despite a substantial percentage of OXPHOS-deficient neurons remaining uncorrected, there was no major encephalopathy phenotype. The protection mechanism given by the minority of corrected neurons is unknown, but some recent observations offer some possibilities. Transfer of

mitochondria between glia and neurons has been reported (English et al, 2020; Hayakawa et al, 2016; Ren et al, 2022), and if active, such process could provide phenotypic relief to defective neurons, if their numbers are not above a specific threshold. Likewise, although not previously reported, healthy neurons could also exchange mitochondria with defective ones. Alternatively, the reduced inflammation or other metabolic alterations in the neural milieu could provide a neuroprotective environment (Lu et al, 2003; Madhavan et al, 2006, 2008; Ourednik et al, 2002). The translational implications of this novel observation are important as they suggest that humans can have a major benefit from a less than complete OXPHOS rescue gene therapy in the CNS.

Although our work shows successful application of an early intervention strategy, mitochondrial diseases can be diagnosed late due to their genetic and symptomatic heterogeneity. Pediatric onset of disease is typically severe, with accelerated progression, and poor prognosis. For an early intervention, fetal genetic testing (as early as the 12th week for rare genetic diseases) would be required. While controversial, in-utero gene therapy could offer a solution for early-onset mitochondrial diseases (Massaro et al, 2018; Rashnonejad et al, 2019). Additional work is required to determine biochemical threshold, or so-called point of no return, in rescuing the phenotype of a mitochondrial encephalopathy. Our findings suggest that a partial rescue has disproportionally positive effects, increasing the chances of clinical success for mitochondrial encephalopathies.

# Methods

### Reagents and tools table

| Reagent/Resource | Reference or Source | Identifier or Catalog Number |
| --- | --- | --- |
| **Experimental Models** | | |
| *Cox10*^fl/fl-CaMKIIα-Cre^+/− | Diaz et al (2012) | N/A |
| *Ndufs3*^fl/fl-CaMKIIα-Cre^+/− | Peralta et al (2020) | N/A |
| **Recombinant DNA** | | |
| pAAV-hSYN-eGFP | Addgene | 50465 |
| pAAV-hSYN-Ndufs3 | This study | N/A |
| pAAV-hSYN-Cox10 | This study | N/A |
| **Antibodies** | | |
| NDUFS3 | Abcam | ab14711 |
| NDUFA9 | Abcam | ab14713 |
| NDUFB8 | Abcam | ab110242 |
| MTCO1 | Abcam | ab14705 |
| COX IV | Abcam | ab14744 |
| SDHA | Abcam | ab14715 |
| UQCRC1 | Abcam | ab110252 |
| VDAC | Abcam | ab14734 |
| GFAP | Cell Signaling | 3680 |
| | Proteintech | 16825-1-AP |
| TUJI | Abcam | ab18207 |

| Reagent/Resource | Reference or Source | Identifier or Catalog Number |
| --- | --- | --- |
| IBA1 | Wako (WB) | 016-20001 |
| | Wako (ICC) | 019-19741 |
| GFP | UC Davis | 75-131 |
| NeuN | Cell Signaling | 12943 |
| Anti-mouse Alexa 488 | Invitrogen | A11001 |
| Anti-rabbit Alexa 568 | Invitrogen | A11036 |
| Anti-mouse Alexa 594 | Invitrogen | A11032 |
| Anti-rabbit Alexa 647 | Invitrogen | A31573 |
| Anti-mouse IgG | Cell Signaling | 7076 |
| Anti-rabbit IgG | Cell Signaling | 7074 |
| **Oligonucleotides and other sequence-based reagents** | | |
| Genotyping primers | | Appendix Table S1 |
| dPCR primer/probe sets | | Appendix Table S2 |
| **Chemicals, Enzymes and other reagents** | | |
| Paraformaldehyde (PFA) | Sigma | 441244 |
| Sucrose | Sigma | S-0389 |
| Tissue-Tek OCT | Sakura | 4583 |
| OmniPur 10X PBS | Calbiotech | 6505 |
| Triton X-100 | Sigma | T9284 |
| Bovine Serum Albumin (BSA) | Sigma | A7030 |
| Vectashield | Biotium | 23004 |
| cOmplete™ Protease Inhibitor Cocktail | Roche | 04693116001 |
| PhosSTOP™ | Roche | 04906845001 |
| Trizma base | Sigma | T1503 |
| Glycine | Sigma | 1.00590 |
| Sodium dodecyl sulfate (SDS) | Sigma | L3771 |
| Trans-Blot Turbo 5X Transfer Buffer | Bio-Rad | 10026938 |
| Ponceau S Solution | Sigma | P7170-1L |
| Tween 20 | Sigma | P1379-500ML |
| SuperSignal Pico West | ThermoFisher Scientific | 34578 |
| Digitonin | Sigma | 300410 |
| 20X Running Buffer | Invitrogen | BN2001 |
| 20X Cathode Additive | Invitrogen | BN2002 |
| NADH | Calbiotech | 481913 |
| Nitrobluetetrazolium | Sigma | N6876 |
| DAB | ThermoScientific | 34001 |
| catalase | Sigma | C9322 |
| Cytochrome c | Sigma | 2506 |
| Proteinase K | Sigma | P2308 |
| **Software** | | |

| Reagent/Resource | Reference or Source | Identifier or Catalog Number |
|---|---|---|
| Activity Monitor | https://med-associates.com/product/activity-monitor-7-software/ | N/A |
| ImageLab | https://www.bio-rad.com/en-us/product/image-lab-software?ID=KRE6P5E8Z | N/A |
| FIJI | https://imagej.net/software/fiji/ | N/A |
| QuPath | https://qupath.github.io/ | N/A |
| Prism | https://www.graphpad.com/ | N/A |
| **Other** | | |
| 5X In-Fusion Snap Assembly Master Mix | Takada | ST2320 |
| One-Shot Stbl3 Chemically Competent E. coli | Invitrogen | C737303 |
| NucleoBond Xtra Maxi EF | Macherey-Nagel | 740424 |
| DC Protein Assay | Bio-Rad | 5000113, 5000114 |
| Mini-PROTEAN TGX Stain-Free 4–20% gel, 10-well | Bio-Rad | 4568094 |
| 0.2 µm PVDF membrane | Bio-Rad | 10026934 |
| 0.2 µM nitrocellulose membrane | Bio-Rad | 162-0097 |
| Mini-PROTEAN Tetra Cell | Bio-Rad | 1658004EDU |
| NativePAGE 3–12% Bis-Tris gel, 10 well | Invitrogen | BN2011BX10 |
| XCell SureLock Mini-Cell | Invitrogen | EI0001 |
| QIAcuity Probe PCR kit | Qiagen | 250101 |
| QIAcuity NanoPlate 8.5k 24-well plate | Qiagen | 250011 |
| Open Field Analysis | Med Associates Inc | N/A |
| RotaRod | Ugo Basile | 47650 |
| Cryostat | Leica | CM 1850-3-1 |
| Synergy H1 Hybrid Reader | Biotek | N/A |
| Trans-Blot Turbo Transfer System | Bio-Rad | 1704150 |
| ChemiDoc Touch Imaging System | Bio-Rad | 1708370 |
| QIAcuity One, 5plex | Qiagen | 911020 |

## Animals

*Ndufs3*-nKO and *Cox10*-nKO mice were previously described in Peralta et al (2020) and Diaz et al (2012), respectively. Briefly, we generated neuron-specific KO mice by mating *Ndufs3*$^{fl/fl}$ or *Cox10*$^{fl/fl}$ males with respective heterozygous, *CaMKIIα*-Cre positive females. When possible, controls and conditional KO mice were obtained from

the same litters. The presence of the WT, floxed genes, and the Cre transgene was detected by PCR from tail DNA (Appendix Table S2).

Cre recombination efficiency was calculated using a three primer PCR approach. A standard curve was created using different ratios of the purified products of floxed and deleted bands from a *Ndufs3*$^{fl/del}$ sample. 10 ng of homogenate-derived DNA was used for samples.

All animals used in this work had a C57BL/6J background and were backcrossed for at least 10 generations. Mice were housed in a virus antigen-free facility at the University of Miami, Division of Veterinary Resources, in a 12-h-light/dark cycle at room temperature and fed ad libitum.

## Adeno-associated viral (AAV) vector production and injection

*Ndufs3* (Gene ID: 68349) and *Cox10* (Gene ID: 70383) cDNA were cloned into an AAV-PHP.eb plasmid containing the hSYN promoter via In-Fusion cloning (Takara). Plasmids were transformed into Stbl3 chemically competent E. coli (Invitrogen, C737303) and prepped with the NucleoBond Xtra Maxi EF kit (Macherey-Nagel). The plasmids were sent to the University of Iowa Viral Core Facility, which produced virus with $4.07 \times 10^{12}$ vg/mL and $6.65 \times 10^{12}$ vg/mL for *Ndufs3* and *Cox10* viruses, respectively. The AAV-*eGFP* was obtained from Penn Vector Core, containing the same promoter and backbone, at a concentration of $1.46 \times 10^{14}$ vg/mL. Juvenile mice (2.5–3 weeks of age) were injected retro-orbitally as previous described (Bacman et al, 2018; Yardeni et al, 2011). *Ndufs3*-nKO mice received $2 \times 10^{11}$ vg of either AAV-*Ndufs3* or AAV-*eGFP*. *Cox10*-nKO mice received $3.32 \times 10^{11}$ vg of either AAV-*Cox10* or AAV-*eGFP*. Considering their average weight at this age, the doses were $2.8 \times 10^{13}$ vg/kg for Ndufs3-nKO and $3.87 \times 10^{13}$ vg/kg for Cox10-nKO mice. Injections were performed retro-orbitally, which immediately drain into the venous system (Yardeni et al, 2011). Retro-orbital injections are more consistent than tail vein injection as the latter tends to collapse if the injection is not precise.

All animal procedures were approved by the University of Miami Animal Care and Use Committee.

## Open field and rotarod testing

Open field (Med Associates Inc.) is a sensitive method for measuring gross and fine locomotor activity. It consists of a chamber and a system of 16 infrared transmitters that record the position of the animal in the 3D space. This system records both horizontal and vertical movement. For our study, the animals were placed in the chamber and the locomotor activities were recorded for 15 min.

Mouse motor coordination was tested at different ages using a Rotarod (Ugo Basile 47650) set at a constant speed of 10 rpm over 180 s. The test consisted of 3 trials performed for each animal at the corresponding age, and the latency to fall was recorded. Mice that completed the task received a final latency time of 180 s. Animals were trained in the rotarod twice per trial, before their first test.

## Tissue staining and microscopy

Anesthetized mice were transcardially perfused with PBS. The brains were isolated, and cortex and hippocampus were dissected from one hemisphere for protein extraction. The second hemisphere was fixed O.N. in 4% PFA, cryoprotected in 30% sucrose,

and frozen in OCT. The brains were cut at a 20 µm thickness with a cryostat and stored at −80 °C (Leica).

Antigen retrieval was performed using a 10 mM sodium citrate buffer (pH 6.0) preheated to 80 °C in a water bath for 30 min. Slides were washed three times in PBS. Sections were permeabilized with 1% Triton, blocked with 2% BSA for 1 h at room temperature, and incubated with primary antibody diluted in 1% BSA and 0.1% Triton overnight at 4 °C. Slides were incubated with Alexa Fluor secondary antibody for 1 h at room temperature and mounted with Vectashield (Biotium 23004). Images were captured with a Leica confocal microscope.

QuPath Cell Detection software was used to detect and count NeuN and GFP-positive cells. For NDUFS3- and COX1-positive NeuN calculations, 3–5 images, captured at 63× with a pixel size of 180 nm, for an average of 200 neurons, were used per mouse. For GFP-postiive NeuN calculations, 3–5 images, captured at 20× with a pixel size of 500 nm, were used per mouse.

| Antibody | Company | Catalog # | Dilution | Species |
|---|---|---|---|---|
| NDUFS3 | Abcam | ab14711 | 1:500 | Mouse |
| MTCO1 | Abcam | ab14705 | 1:100 | Mouse |
| GFAP | Cell Signaling | 3680 | 1:500 | Mouse |
|  | Proteintech | 16825-1-AP | 1:500 | Rabbit |
| NeuN | Cell Signaling | 12943 | 1:500 | Rabbit |
| IBA1 | Wako | 019-19741 | 1:1000 | Rabbit |
| GFP | UC Davis | 75-131 | 1:500 | Mouse |
| Alexa 488 | Invitrogen | A11001 | 1:500 | Mouse |
| Alexa 568 | Invitrogen | A11036 | 1:500 | Rabbit |
| Alexa 594 | Invitrogen | A11032 | 1:500 | Mouse |
| Alexa 647 | Invitrogen | A31573 | 1:500 | Rabbit |

## Western blotting

Protein extracts were prepared from the cortex and hippocampus regions and homogenized in PBS containing a protease and phosphatase inhibitor mixture (Roche). Samples were sonicated for ~5 s. Protein concentration was determined by the Lowry assay using the DC kit (Bio-Rad). Approximately 20–50 µg protein was separated by SDS-PAGE in 4–20% acrylamide gels (Bio-Rad). Gels were run in Tris-glycine buffer (25 mM Tris-Cl, 250 mM glycine, 0.1% SDS) at 100 V until completion. Gels were transferred to 0.2 µm PVDF membranes (Bio-Rad) with Trans-Blot Turbo Transfer System and 1X Trans-Blot Turbo Transfer Buffer (Bio-Rad), and imaged after for total protein loading. For IBA1, gels were transferred to 0.2 µm nitrocellulose membranes, stained with Ponceau for total protein loading, and otherwise treated the same as PVDF membranes.

Membranes were blocked with 5% nonfat milk in 0.1% Tween-20 in PBS and subsequently incubated with specific antibodies, which were incubated overnight at 4 °C or 1 h at room temperature. Blots were incubated with secondary antibodies conjugated to horseradish peroxidase (Cell Signaling Technology) for 1 h at room temperature. The reaction was developed by chemiluminescence using SuperSignal Pico West reagent (Thermo Fisher Scientific, #34578). Blots were visualized with ChemiDoc Imaging System (Bio-Rad). Optical density measurements were taken by software supplied by Bio-Rad. Total protein loading (TPL) is used as control. Blots and their respective gel staining can be found in Appendix Figure S5.

| Antibody | Company | Catalog # | Dilution | Species |
|---|---|---|---|---|
| NDUFS3 | Abcam | Ab14711 | 1:1000 | Mouse |
| NDUFB8 | Abcam | Ab110242 | 1:1000 | Mouse |
| MTCO1 | Abcam | Ab14705 | 1:1000 | Mouse |
| UQCRC1 | Abcam | Ab110252 | 1:1000 | Mouse |
| SDHA | Abcam | Ab14715 | 1:1000 | Mouse |
| IBA1 | Wako | 016-20001 | 1:500 | Rabbit |
| GFAP | Proteintech | 16825-1-AP | 1:4000 | Rabbit |
| TUJI | Abcam | Ab18207 | 1:20,000 | Rabbit |
| GFP | UC Davis | 75-131 | 1:1000 | Mouse |
| Anti-mouse IgG | Cell Signaling | 7076 | 1:5000 | Horse |
| Anti-rabbit IgG | Cell Signaling | 7074 | 1:5000 | Goat |

## BN-PAGE and in-gel activity assay

Cortex homogenates were prepared by centrifuging at $10,000 \times g$ for five minutes to isolate a purer mitochondrial fraction. Pellets were resuspended in a 1.5 M aminocaproic acid, 50 mM Tris-HCl buffer. Homogenates were solubilized with 4 mg of digitonin (Sigma, #300410) per 1 mg of protein. To determine the levels of isolated respiratory complexes and supercomplexes, 50 µg of protein was separated on by BN-PAGE in 3–12% acrylamide gradient gels (Invitrogen) in xCell *SureLock* Mini-Cell tank containing 1X Running Buffer (Invitrogen, BN2001). Gels were run in Dark Cathode buffer (1X Running Buffer with 5% Cathode Buffer Additive (Invitrogen, BN2002)) for 30–45 min at 150 V. Dark Cathode Buffer was exchanged for Light Cathode Buffer (1X Running Buffer with 0.5% Cathode Buffer Additive) and run until completion at 220 V. Gels were transferred to PVDF membrane (Bio-Rad) via wet transfer in bicarbonate buffer (10 mM sodium bicarbonate, 3 mM sodium carbonate), for 1 h at a constant current of 300 mA. Blots were blocked in 5% milk in 0.1% Tween-20 in PBS and incubated with antibodies against several subunits of mitochondrial respiratory complexes overnight at 4 °C. Blots were incubated with secondary antibody conjugated to horseradish peroxidase (Cell Signaling Technology) for 1 h at room temperature. The reaction was developed by chemiluminescence using SuperSignal Pico West reagent (Thermo Fisher Scientific, #34578). Blots were visualized with ChemiDoc Imaging System (Bio-Rad). Optical density measurements were taken by software supplied by Bio-Rad. CII band density is used as control.

| Antibody | Company | Catalog # | Dilution | Species |
|---|---|---|---|---|
| NDUFA9 | Abcam | Ab14713 | 1:1000 | Mouse |
| MTCO1 | Abcam | Ab14705 | 1:1000 | Mouse |
| UQCRC1 | Abcam | Ab110252 | 1:1000 | Mouse |
| SDHA | Abcam | Ab14715 | 1:1000 | Mouse |
| Anti-mouse IgG | Cell Signaling | 7076 | 1:5000 | Horse |

To detect the activity of Complex I in gel, mitochondrial complexes treated with digitonin and separated in 3–12% gels were incubated overnight at room temp, in a buffer of 14 mM NADH (Calbiotech, #481913), 1 mg/mL nitroblue tetrazolium (Sigma, N6876), and 5 mM Tris-HCl pH 7.4. To detect the activity of Complex IV in gel, the above-described gels were incubated overnight at room temp, in a buffer of 50 mM Potassium Phosphate, 1 mg/mL DAB (ThermoScientific, #34001), 24 U/mL catalase (Sigma, #C9322), 1 mg/mL cytochrome c (Sigma, #C2506), and 75 mg/mL sucrose. Optical density measurements were taken in FIJI. CII band density was used as control.

### DNA extraction and viral titer quantification

Genomic DNA was extracted from cortex and hippocampal tissue using standard proteinase K, phenol, chloroform extraction, and isopropyl alcohol precipitation. Digital PCR reactions used IDT custom primer and probe sets (Appendix Table S3) in combination with QIAcuity Probe PCR kit (Qiagen) and were performed on QIAcuity One (Qiagen). The mtDNA/nDNA ration was determined using 1 ng of genomic DNA and probes for ND1 to 18S. The AAV viral titer was estimated using 50 ng of genomic DNA and the ratio of hSYN to TTR.

### Statistics and reproducibility

Sample sizes were determined based on previous publications, and independent biological replicates range from 3 to 4 for all experimental modalities used in this study. No data were excluded from the analysis. A few replicates are missing due to a failure of

**The paper explained**

**Problem**

Mitochondrial diseases are a heterogenous group of genetic diseases caused by mutations in nuclear DNA or mitochondrial DNA (mtDNA), ultimately affecting the oxidative phosphorylation (OXPHOS) system. These diseases are often multisystemic, but the CNS is frequently affected. Approximately 1 in 5000 people worldwide suffer from mitochondrial disease, for which no cures or effective treatments exist. This study investigated whether missing nuclear genes coding for OXPHOS proteins could be replaced in neurons.

**Results**

Using two mouse models of mitochondrial encephalopathy, featuring deficiencies in Complex I (*Ndufs3*-nKO) or Complex IV (*Cox10*-nKO), we performed gene replacement therapy via AAV-PHP.eB in juvenile, pre-symptomatic mice. The treatments were able to block disease onset. Surprisingly, we found that despite only a fraction of neurons receiving the virus, OXPHOS activity was restored to levels that were enough to reduce neuroinflammation and prevent an encephalopathy phenotype.

**Impact**

Here we showed the successful application of an early gene therapy intervention strategy in two models of mitochondrial encephalopathy. We found that a partial rescue had disproportionally positive effects. Our work suggests that favorable clinical outcomes could be obtained in patients exhibiting OXPHOS deficiencies, even with incomplete gene delivery.

acquisition of the image after dPCR. The experiments were not randomized. The investigators were not blinded to allocation during experiments and outcome assessment. Analysis was performed with Prism GraphPad. Exact *p*-values can be found in Appendix Table S1.

## Data availability

This study includes no data deposited in external repositories.

The source data of this paper are collected in the following database record: biostudies:S-SCDT-10_1038-S44321-024-00111-4.

## Peer review information

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

## Acknowledgements

The authors also thank the members of the Moraes Lab for constant intellectual input. This work was funded primarily by the National Institute of Health (NIH) Grant 5R01EY010804, the Florida Biomedical Foundation (21K05) and the Army Research Office (W911NF-21-1-0248), with secondary support from NIH/NINDS 1R01NS079965, the Muscular Dystrophy Association (MDA 964119), and the Research to Prevent Blindness (RPB-23 Stein).

## Author contributions

**Brittni R Walker**: Conceptualization; Data curation; Formal analysis; Investigation; Methodology; Writing—original draft; Writing—review and editing. **Lise-Michelle Theard**: Investigation; Writing—review and editing. **Milena Pinto**: Investigation; Methodology; Writing—review and editing. **Monica Rodriguez-Silva**: Investigation; Writing—review and editing. **Sandra R Bacman**: Investigation; Methodology; Writing—review and editing. **Carlos T Moraes**: Conceptualization; Supervision; Funding acquisition; Project administration; Writing—review and editing.

Source data underlying figure panels in this paper may have individual authorship assigned. Where available, figure panel/source data authorship is listed in the following database record: biostudies:S-SCDT-10_1038-S44321-024-00111-4.

## Disclosure and competing interests statement

The authors declare no competing interests.

# Expanded View Figures

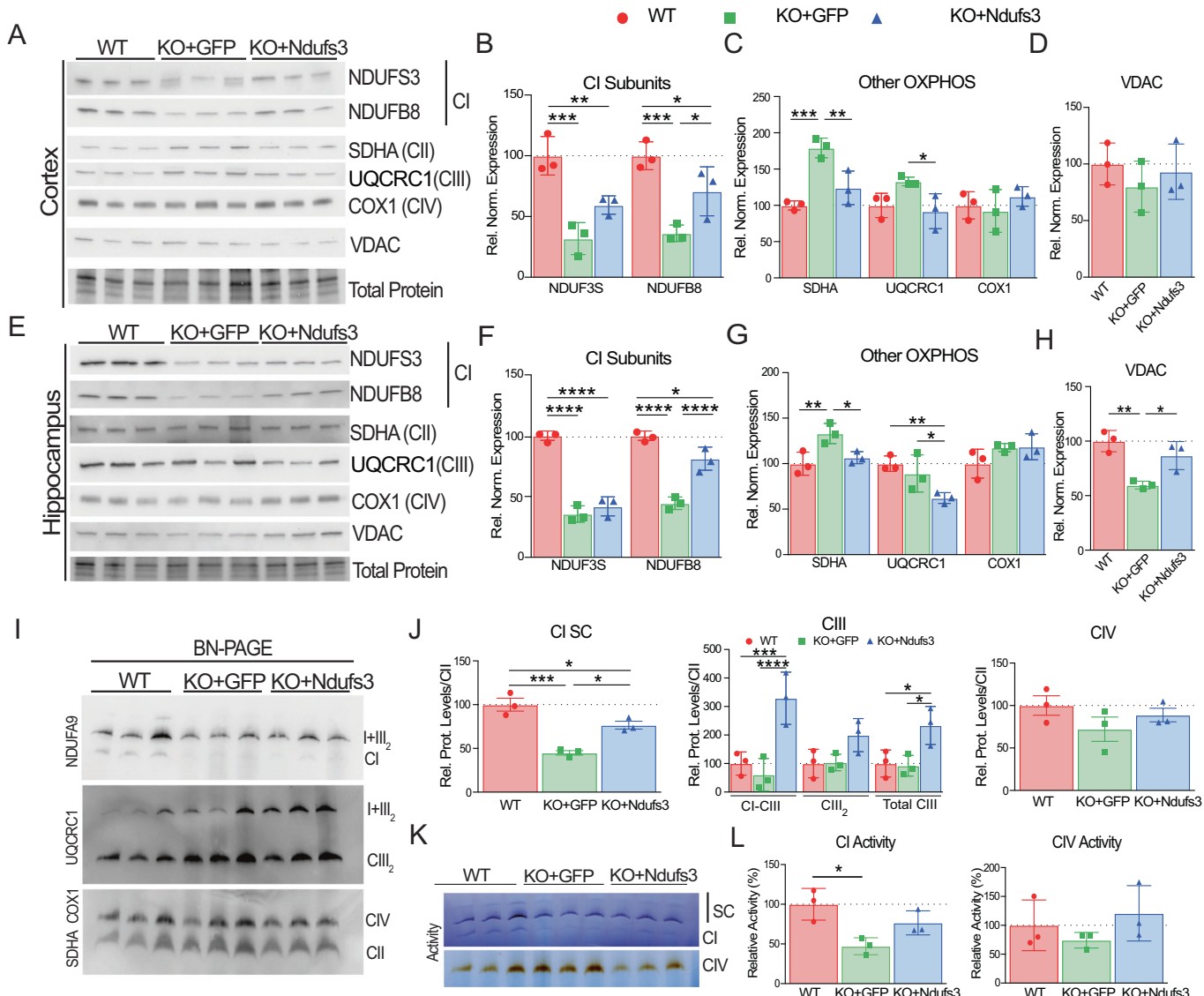

**Figure EV1.    Restoration of NDUFS3 in females *Ndufs3*-nKO mice.**

(A–H) Western blots and relative quantifications of cortical and hippocampal homogenates of 5-month-old wild-type (WT), *Ndufs3*-nKO+GFP (KO + GFP), and *Ndufs3*-nKO+*NdufsS3* (KO+Ndufs3) female mice. Probed for NDUFS3 and NDUFB8 (Complex I subunits), SDHA (Complex II subunit), CORE1 (Complex II subunit, COX1 (Complex IV subunit), and VDAC (mitochondrial membrane protein). Total protein loading was used as loading control. All protein loading staining and their respective blots are shown in Appendix Fig. S5. (I, J) BN-PAGE and relative quantifications of steady-state levels of respiratory complexes normalized to CII levels. (K, L) BN-PAGE in gel activity and relative quantifications of enzymatic activity. Data information: In (B, C, F, G, J (CIII)), data are represented as mean ± SD (*n* = 3/group). *P* values were calculated using two-way ANOVA, with Tukey's multiple comparisons test. In (D, H, J, L), data are represented as mean ± SD (*n* = 3/group). *P* values were determined by one-way ANOVA, with Tukey's multiple comparisons test. *P*(*) = 0.0332, *P*(**) = 0.0021, *P*(***) = 0.0002, *P*(****) < 0.0001. Exact *P* values are listed in Appendix Table S1. Source data are available online for this figure.

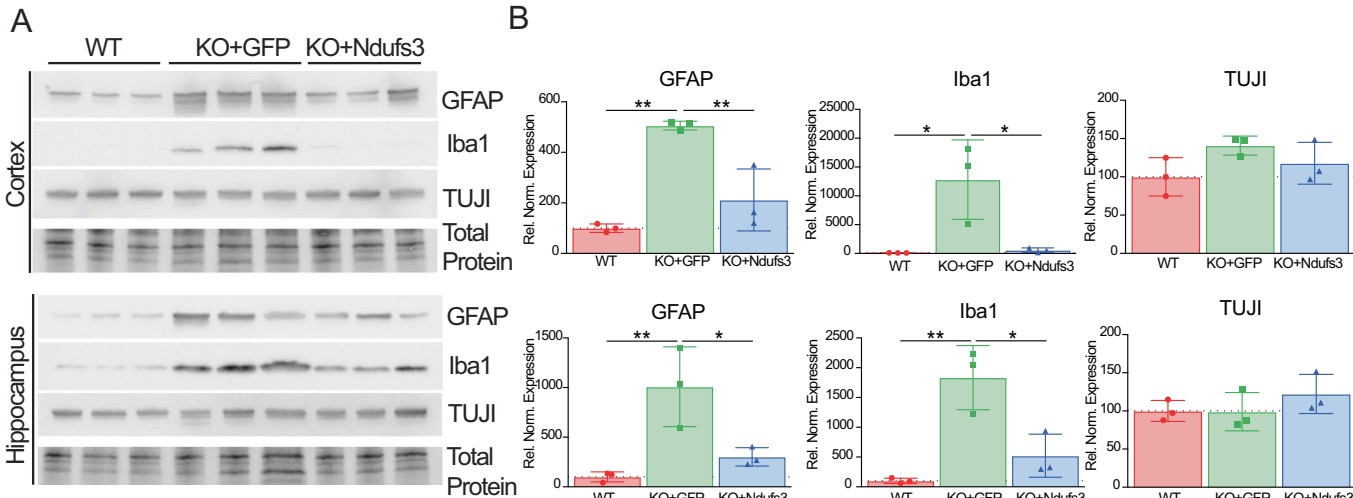

**Figure EV2.  Figure 6. Neuroinflammation in *Ndufs3*-nKO female mice is improved by CNS gene therapy.**

(A, B) Western blots and relative quantification of protein homogenates from cortex and hippocampus of female WT, KO+*eGFP*, and KO+*Ndufs3* mice at 5 months of age, probing for astrocyte activation (GFAP), microglial marker IBA1, and neuronal marker TUJ1. Total protein loading was used as loading control. All protein loading staining and their respective blots are shown in Appendix Fig. S5. Data Information: In (B), data are represented as mean ± SD (*n* = 3/group). *P* values were calculated using one-way ANOVA, with Tukey's multiple comparisons test. *P*(*) = 0.0332, *P*(**) = 0.0021. Exact *P* values are listed in Appendix Table S1. Source data are available online for this figure.

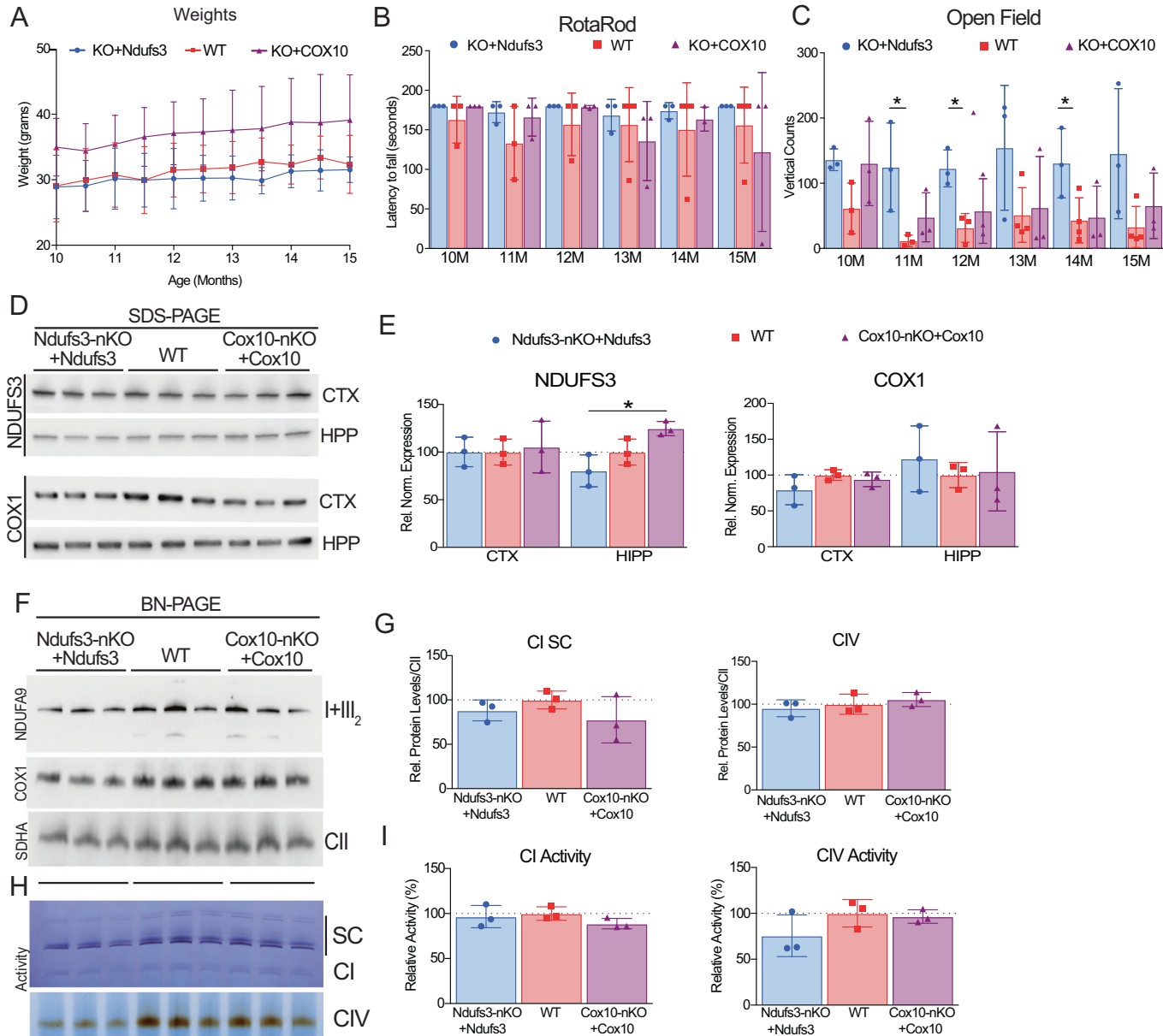

**Figure EV3. Extended survival study.**

(A) Weekly weights of *Ndufs3*-nKO+*NdufsS3* (blue), *Cox10*-nKO+*Cox10* (purple), and WT (red) mice between 10 and 15 months of age. (B) Rotarod performed by *Ndufs3*-nKO+*Ndufs3* (blue), WT (red), and *Cox10*-nKO+*Cox10* (purple) mice at 10, 11, 12, 13, 14, and 15. (C) Vertical counts recorded during open field analysis of *Ndufs3*-nKO+*Ndufs3* (blue), WT (red), and *Cox10*-nKO+*Cox10* (purple) mice at 10, 11, 12, 13, 14, and 15. (D, E) Western blot and relative quantifications of NDUFS3 and COX1 in cortex and hippocampal homogenates of 15-month-old animals. Total protein loading was used as loading control. All protein loading staining and their respective blots are shown in Appendix Fig. S5. (F, G) BN-PAGE and relative quantifications of steady-state levels of respiratory complexes, normalized to CII levels. (H, I) BN-PAGE in gel activity and relative quantifications of enzymatic activity. Data information: In (A, B, C, E), data are represented as mean ± SD (*n* = 3–4/group). *P* values were calculated using two-way ANOVA, with Tukey's multiple comparisons test, compared to WT. In (G, I), data are represented as mean ± SD (*n* = 3/group). *P* values were determined by one-way ANOVA, with Tukey's multiple comparisons test, compared to WT. Exact *P* values are listed in Appendix Table S1. Source data are available online for this figure.

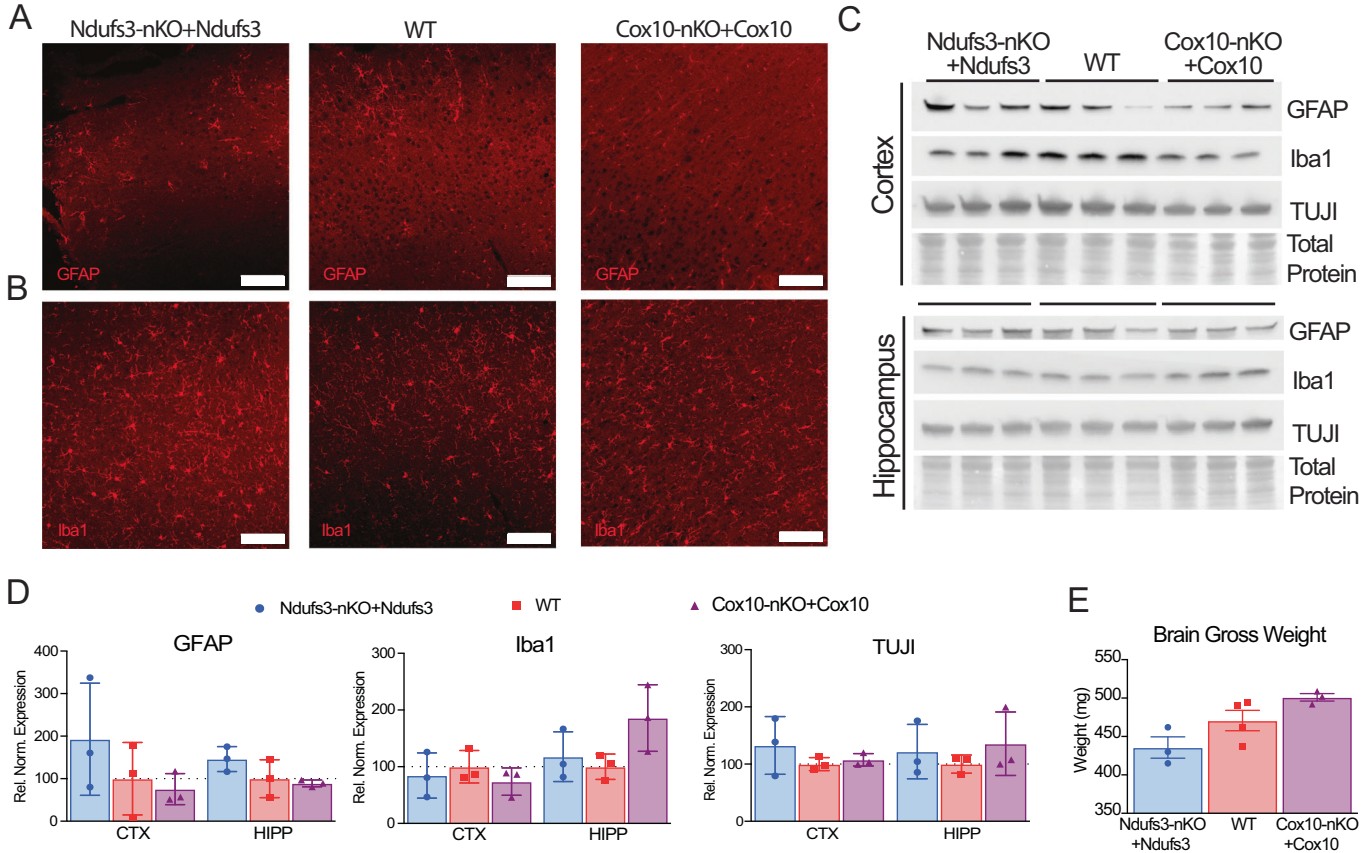

**Figure EV4. Neuropathology of 15-month-old OXPHOS-nKO mice after gene therapy.**

(A) Immunohistochemical images of GFAP in motor cortex of 15-month-old male mice. Scale bar is 100 µm. (B) Immunohistochemical images of IBA1 in motor cortex of 15-month-old male mice. Scale bar is 100 µm. (C, D) Western blots and relative quantification of protein homogenates from cortex and hippocampi of *Ndufs3*-nKO-*Ndufs3*, WT, and *Cox10*-nKO+*Cox10* mice at 15 months of age, probing for GFAP, IBA1, and TUJ1. Total protein loading was used as loading control. All protein loading staining and their respective blots are shown in Appendix Fig. S5. (E) Brain weight of 15-month-old *Ndufs3*-nKO+*Ndufs3* (blue), WT (red), and *Cox10*-nKO+*Cox10* (purple), mixed sex. Data information: In (D), data are represented as mean ± SD ($n = 3$/group). $P$ values were calculated using two-way ANOVA with Tukey's multiple comparisons test, compared to WT. In (E), data are represented as mean ± SD ($n = 3$–4/group). $P$ values were determined by one-way ANOVA, with Tukey's multiple comparisons test, compared to WT. $P(*) = 0.0332$. Exact $P$ values are listed in Appendix Table S1. Source data are available online for this figure.

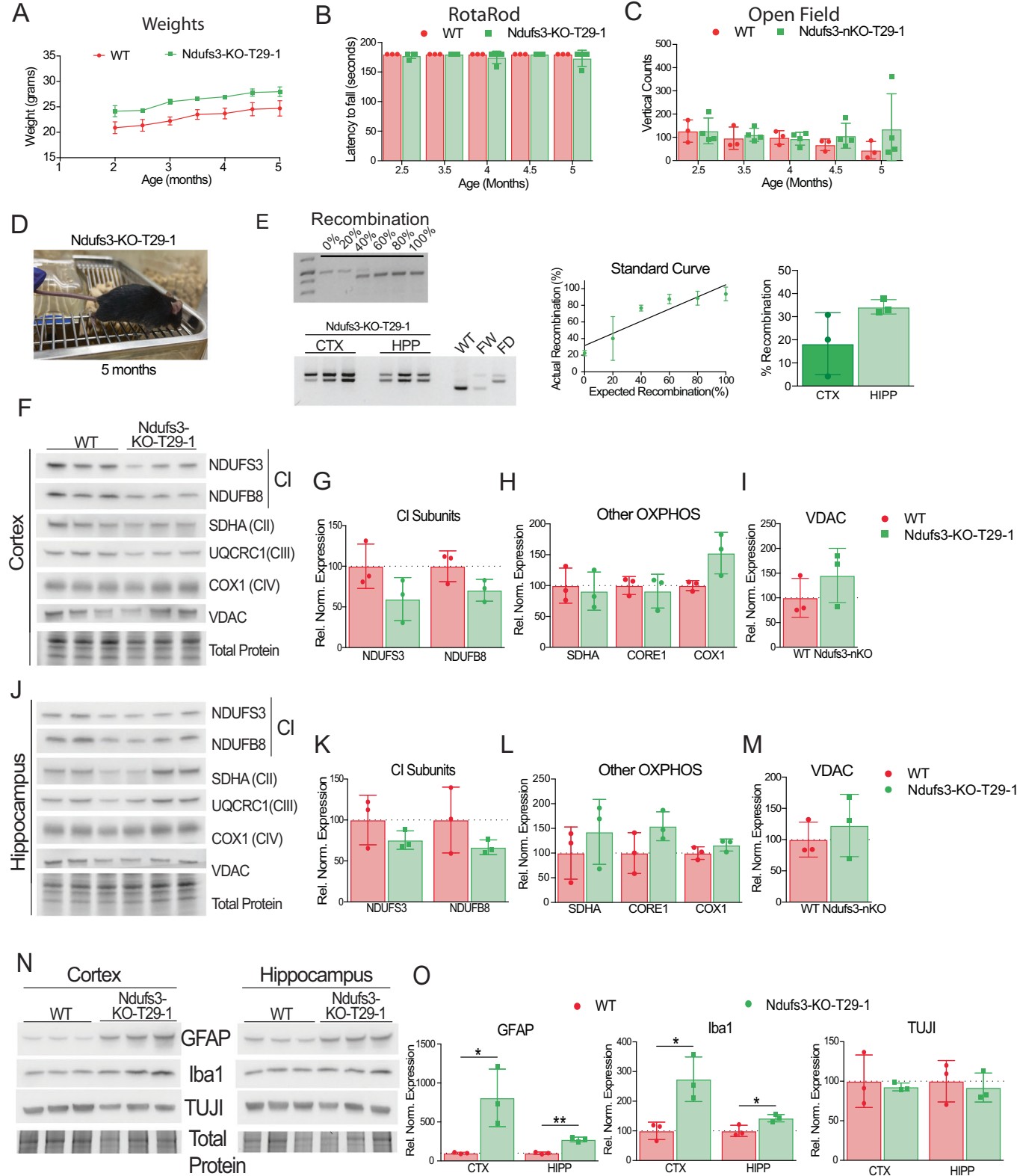

◄ **Figure EV5. T29-1-cre driven CNS deletion of *Ndufs3* does not cause overt CNS phenotypes.**

(**A**) Weekly weights of *Ndufs3*-nKO-T29-1 mice over the course of the age-matched study. (**B**) Rotarod performed by WT (red) and *Ndufs3*-nKO-T29-1 (green) mice at 2.5, 3.5, 4, 4.5, and 5 months of age. (**C**) Vertical counts recorded during open field analysis of *Ndufs3*-nKO-T29-1 mice at 2.5, 3.5, 4, 4.5, and 5 months of age. (**D**) Representative image of 5-month-old *Ndufs3*-nKO-T29-1 mice. (**E**) Representative gel of three primer PCR, showing recombination of *Ndufs3*-nKO-T29-1 mice at 8 months of age. Quantification of the three primer PCR for cortex and hippocampus. (**F–M**) Western blots and relative quantifications of protein homogenates from cortex and hippocampus of 8-month-old wild-type (WT) and *Ndufs3*-nKO-T29-1 male mice probed for NDUFS3 and NDUFB8 (Complex I subunits), SDHA (Complex II subunit), UQCRC1 (Complex III subunit), COX1 (Complex IV subunit), and VDAC (mitochondrial membrane protein). Total protein loading was used as loading control. All protein loading staining and their respective blots are shown in Appendix Fig. S5. (**N, O**) Western blots and relative quantification of protein homogenates from cortex and hippocampus of males WT and Ndufs3-nKO-T29-1 mice at 8 months of age, probing for astrocyte activation (GFAP), microglial marker IBA1, and neuronal marker TUJ1. Total protein loading was used as loading control. All protein loading staining and their respective blots are shown in Appendix Fig. S5. Data information: Data are represented as mean ± SD ($n = 3$/group). $P$ values were calculated using Welch's t-test, or multiple t-tests with Holm Sidak's multiple comparisons test. $P(*) = 0.0332$. Exact $P$ values are listed in Appendix Table S1. Source data are available online for this figure.

