## [Peer Review File · EMBO Molecular Medicine]

Restoration of Defective Oxidative Phosphorylation to a Subset of Neurons Prevents Mitochondrial Encephalopathy

Brittni Walker, Lise-Michelle Theard, Milena Pinto, Monica Rodriguez-Silva, Sandra Bacman, and Carlos Moraes

Corresponding author: Carlos Moraes (CMoraes@med.miami.edu)

Review Timeline:

Submission Date:	17th Apr 24
Editorial Decision:	8th May 24
Revision Received:	7th Jun 24
Editorial Decision:	2nd Jul 24
Revision Received:	14th Jul 24
Accepted:	16th Jul 24

Editor: Zeljko Durdevic

Transaction Report:

8th May 2024

Dear Prof. Moraes,

Thank you for the submission of your manuscript to EMBO Molecular Medicine. We have now received feedback from the three reviewers who agreed to evaluate your manuscript. While the referee #1 is overall supportive, referees #2 and #3 recognize interest of the study but also raise important concerns that should be addressed in a major revision. If you would like to discuss further the points raised by the referees, I am available to do so via email or video. Let me know if you are interested in this option.

We would welcome the submission of a revised version within three months for further consideration. Please let us know if you require longer to complete the revision.

I look forward to receiving your revised manuscript.

Yours sincerely,

Zeljko Durdevic

We require:

- 1) A .docx formatted version of the manuscript text (including legends for main figures, EV figures and tables). Please make sure that the changes are highlighted to be clearly visible.
- 2) Individual production quality figure files as .eps, .tif, .jpg (one file per figure). For guidance, download the 'Figure Guide PDF': (<https://www.embopress.org/page/journal/17574684/authorguide#figureformat>).
- 3) A .docx formatted letter INCLUDING the reviewers' reports and your detailed point-by-point responses to their comments. As part of the EMBO Press transparent editorial process, the point-by-point response is part of the Review Process File (RPF), which will be published alongside your paper.
- 4) A complete author checklist, which you can download from our author guidelines (<https://www.embopress.org/page/journal/17574684/authorguide#submissionofrevisions>). Please insert information in the checklist that is also reflected in the manuscript. The completed author checklist will also be part of the RPF.
- 5) Please note that all corresponding authors are required to supply an ORCID ID for their name upon submission of a revised manuscript.
- 6) It is mandatory to include a 'Data Availability' section after the Materials and Methods. Before submitting your revision, primary

datasets produced in this study need to be deposited in an appropriate public database, and the accession numbers and database listed under 'Data Availability'. Please remember to provide a reviewer password if the datasets are not yet public (see <https://www.embopress.org/page/journal/17574684/authorguide#dataavailability>).

13) Author contributions: You will be asked to provide CRediT (Contributor Role Taxonomy) terms in the submission system. These replace a narrative author contribution section in the manuscript.

14) A Conflict of Interest statement should be provided in the main text.

15) Every published paper now includes a 'Synopsis' to further enhance discoverability. Synopses are displayed on the journal webpage and are freely accessible to all readers. They include a short stand first (maximum of 300 characters, including space) as well as 2-5 one-sentence bullet points that summarize the paper. Please write the bullet points to summarize the key NEW findings. They should be designed to be complementary to the abstract - i.e. not repeat the same text. We encourage inclusion of key acronyms and quantitative information (maximum of 30 words / bullet point). Please use the passive voice. Please attach these in a separate file or send them by email, we will incorporate them accordingly.

Please also suggest a striking image or visual abstract to illustrate your article as a PNG file 550 px wide x 300-800 px high.

**** Reviewer's comments ****

Referee #1 (Comments on Novelty/Model System for Author):

The chosen models for mitochondrial disease are mice conditional knockouts. Mice models of mito disease often do not display similar phenotypes to the human ones but in these cases the models do display encephalopathy and it is that presentation that is being rescued.

Referee #1 (Remarks for Author):

The manuscript from Walker and colleagues reports the rescue of the neuron-specific deletion of NDUFS2 and COX10 by adeno-associated viral expression of the respective wild type genes in the CNS. The paper is well written, concise and clear. It is notable that this expression can rescue the encephalopathy associated with either deletion as well as rescue of OXPHOS and, partially, neuroinflammation. Perhaps the most important message from this impressive work is that rescue appears to only be required in a subset (~30%) of neurons. The authors suggest this data is consistent with the defective neurons being protected by the local environment and give several possibilities as to how this may occur. Overall, this is a thorough and impressive piece of work and I recommend publication. I have a couple of minor points that the authors may wish to address:

1. I Recommend that the authors briefly explain why retro-orbital injection is used.
2. Fig 5E,F I guess the levels of CI in F is not related to the BN gels in E, or are they? Irrespective, how were the steady state levels calculated? Why do the authors think the steady state level of CI is low in the COX10 KO when the activity is actually greater than wild type?
3. 'IBA1 staining revealed similar levels of microglia (Fig. 6B).' I found this a confusing statement. Similar to what? I think the authors meant that the microglia staining with IBA1 was also increased back to wild type levels in the rescued mice. Please clarify.

Referee #2 (Remarks for Author):

This is an interesting manuscript in which gene therapy is used to restore mitochondrial function to a subset of affected neurons. Mitochondrial function had been compromised via 2 mouse models, a complex I mutant subunit and a complex IV mutant assembly factor. Both models are well established. There is clear evidence for effective gene therapy and thus this paper is of considerable medical interest.

Despite its interest, the work reported is largely observational and would benefit from mechanistic enhancement. Some specific points are:

1. There are a number of sex-specific effects. For example, reintroduction of NDUFS3 in females (Fig. S1B) was not as successful as that observed in Fig. 3, yet there was a rescue of the phenotype. Also, mitochondrial mass (VDAC) is increased 30-40% (Fig. 3D) on reintroduction of NDUFS3 in males but trends lower in females (S1D). Can the authors clarify this?
2. Total protein was used to normalize the protein levels in immunoblots. The authors have selected one specific size for the normalization. Also, there are multiple bands shown in each figure. Could the authors provide details of which specific band was used to quantify and the rationale? Why was total protein better than the usual housekeeping markers?

3. Since the end product of OxPhos is ATP, it's surprising that mitochondrial ATP levels are not shown. Inclusion of ATP levels after gene reintroduction may help understand the individual complex levels and activities.
4. Some results are not consistent with previous observations. For example, in their Peralta et al. paper (JCI), COX I shows a 3-fold increase at 4 mo in their Ndufs3 KO whereas in the current manuscript there is no increase (Fig. 3c).
5. A result like that in Fig. 4J showing about 70% overlap of COX1 and NeuN - does this mean that some neurons don't contain COX1 or is there a technical limitation? Also, what is the meaning of a result like ~30% COX1+NeuN in that figure or about 40% NDUFS3 expression in Fig. 3B? Is the knockout incomplete? And why is the level at 5 months so much higher than the ~20% level at only 4 months in the JCI paper?

Referee #3 (Remarks for Author):

The manuscript entitled "Restoration of Defective Oxidative Phosphorylation to a Subset of Neurons Prevents Mitochondrial Encephalopathy" describes a study in which gene replacement approaches are used to treat two distinct mouse models of mitochondrial encephalopathy. In general, the results are promising as each disease mouse model does appear to benefit from their respective treatment strategy. My major concerns are related to the presentation of male/female data - this may impact effect sizes as your cohort numbers are low. Also, in several instances throughout the manuscript a significant improvement is declared due to differences between affected and affected-treated mice despite there being no difference between WT healthy controls and affected mice to begin with. My specific comments are as follows:

- 1) Throughout the manuscript, at times male and female data are combined and at other times they are separated. Based upon your data, sex appears to be a significant biological variable. Male/female data should be presented separately (different bars) but displayed in the same graphs in the main figures of the manuscript. Exact numbers of males and females included per cohort or experiment need to be indicated in figure legends.
- 2) Present doses in vg/kg.
- 3) Figure 2B - Asterisk markings are not clearly positioned - significance appears to be between treated and GFP-treated only - not between GFP-treated and WT. Males and females should be separated for all data presented. Exact number included per cohort needs to be indicated.
- 4) Figure 2E - As open field tests apparently show no significant differences between WT and affected mice at any age, this is not very interesting and could be supplemental data.
- 5) Page 6 - 1st paragraph - It is an overstatement to say that Ndufs-nKO mice show a trend in increase in mtDNA copy numbers based upon the data presented. Also, gene replacement does not appear to have any significant effect on this.
- 6) The mouse cohort numbers used throughout this manuscript are very low. Significant differences are the only ones that should be called out - there are many instances where things are described as elevated or increased when the data show no significant differences. This needs to be corrected throughout the manuscript - almost every figure. The data that show no significant differences between groups can be moved to supplemental data.
- 7) Cox10-nKO female treated/untreated data need to be provided.
- 8) Figure 6 - female data need to be provided and presented alongside the male data.
- 9) Figure 7 - female data need to be provided and presented alongside the male data.
- 10) Figure S2 - males and females are not indicated and should be presented separately.
- 11) Based upon Figure 7 data it seems there may be a loss of NeuN+ cells over time in KO-GFP mouse brains. Unclear if this is true for both models or only one but loss of a cell type does sometimes result in altered AAV vg persistence in tissues.
- 12) Figure 8 should be supplemental data and should include images from both models.
- 13) Some supplemental figures and legends need editing. S1 title says female mice but the legend says male mice. S3 title only mentions one model not both.
- 14) Some discussion on what other cell types may have been transduced by this vector is needed.
- 15) The discussion over-states the results in many cases as the treatment effects in females are not as significant or were not presented.
- 16) Some discussion needed for why are there differences between the treated and affected mice when there were no differences between WT and affected mice for many of the assessments presented.
- 17) Some discussion needed regarding differences between cell type affected in the models used (only neurons) and those impacted in the human clinical presentation of these disorders. These are good mouse models but residual functional proteins in surrounding non-neurons or potential for overexpression from AAV transduction of other non-neuron cells may have an impact on the mouse phenotypes and can be discussed.

We would like to thank the reviewers for their constructive suggestions and corrections. Below, we address the specific points raised.

Referee #1

1) I recommend that the authors briefly explain why retro-orbital injection is used.

Retro-orbital injection, an established delivery method, was used for intra-venous administration due to the size of the mice. Although tail vein administration allows for more volume to be injected, it can be difficult to perform in small mice, causing distress to the mice, and has a high rate of failure. Our lab, as well as several others, has successfully utilized retro-orbital injections for multiple mouse models and viral constructs. A review from 2001, describes the method and highlights its advantages (Yardeni *et al*, 2011). We added this explanation to the Methods (page 15, line 12).

2) Fig 5E,F I guess the levels of CI in F is not related to the BN gels in E, or are they ?

Irrespective, how were the steady state levels calculated? Why do the authors think the steady state level of CI is low in the COX10 KO when the activity is actually greater than wild type ? The Complex I steady state levels were calculated via densitometric measurements of the CI+CIII supercomplex band and normalized to the CII band of the same blot (using a Chemidoc detection system, BioRad). Activity levels were calculated in a similar manner and normalized to the CII band of the blot run with the sample samples. After reviewing the Reviewers comments, we re-analyzed the activity gel and noticed a very uneven background. We re-analyzed the gel using better background correction and confirmed the visual inspection that KO+GFP mice had decreased CI activity, which is also in agreement with the CI SC steady state levels. Figure 5H was revised accordingly.

3) IBA1 staining revealed similar levels of microglia (Fig. 6B). I found this a confusing statement. Similar to what ? I think the authors meant that the microglia staining with IBA1 was also increased back to wild type levels in the rescued mice. Please clarify.

Our wording was unclear regarding the IBA1 staining. We have corrected the text to state that the pattern was similar to the GFAP pattern (i.e. increased in the KO and normalized in the treated). Page 8, line 12-13.

Referee #2

1. There are a number of sex-specific effects. For example, reintroduction of NDUFS3 in females (Fig. S1B) was not as successful as that observed in Fig. 3, yet there was a rescue of the phenotype. Also, mitochondrial mass (VDAC) is increased 30-40% (Fig. 3D) on reintroduction of NDUFS3 in males but trends lower in females (S1D). Can the authors clarify this?

Although *Ndufs3*-nKO+*Ndufs3* females did not have as great of a molecular recovery of NDUFS3 (by western blots), we did see an approximately 30% increase in Complex I supercomplex assembly and activity. Besides the fact that western blots are semi quantitative assays, this observation further supports our later point that a relatively small percentage of neurons transduced can have a disproportional beneficial phenotypic effect. It is unclear why

VDAC trends lower in female *Ndufs3*-nKO+*eGFP*. Mouse data can be variable, and the small differences between males and females for *Ndufs3* and *VDAC1* may not be sex-related.

2. Total protein was used to normalize the protein levels in immunoblots. The authors have selected one specific size for the normalization. Also, there are multiple bands shown in each figure. Could the authors provide details of which specific band was used to quantify and the rationale? Why was total protein better than the usual housekeeping markers?

Regarding total protein, the entire lane was quantified and used for normalization; however, to have a more proportional figure panel, we cropped the image to show the 25-35kDa range, where our two proteins of interest lie. There are different views on what the best loading controls are for western blots, but some housekeeping proteins, such as β -actin, GAPDH, and α -tubulin have been shown to not be good loading controls for neuropathological samples, as they can change in response to different stresses (Goasdoue *et al*, 2016). Other studies showed that beta-actin is not a reliable marker (Dittmer & Dittmer, 2006). Total protein normalization is arguably more accurate as it includes a large number of endogenous proteins (Aldridge *et al*, 2008).

3. Since the end product of OxPhos is ATP, it's surprising that mitochondrial ATP levels are not shown. Inclusion of ATP levels after gene reintroduction may help understand the individual complex levels and activities.

Although we agree with the Reviewer that data on ATP levels would be interesting, ATP assays are more accurate when using fresh or freshly frozen samples. Moreover, ATP levels generated from glial cells, which contain a large amount of glycolytic astrocytes (Takahashi, 2021) and oligodendrocytes (Funfschilling *et al*, 2012), could mask differences in homogenates.

4. Some results are not consistent with previous observations. For example, in their Peralta et al. paper (JCI), COX I shows a 3-fold increase at 4 mo in their *Ndufs3* KO whereas in the current manuscript there is no increase (Fig. 3c).

We have noted some small differences in COX1 levels between this group of *Ndufs3*-nKO mice when compared to our previous paper; however, in Peralta et al. we normalized protein levels to β -actin, whereas we now normalized western bands to total protein loading. Additionally, we now analyzed mice at a slightly later time point, which may affect these changes.

5. A result like that in Fig. 4J showing about 70% overlap of COX1 and NeuN - does this mean that some neurons don't contain COX1 or is there a technical limitation? Also, what is the meaning of a result like ~30% COX1+NeuN in that figure or about 40% *NDUFS3* expression in Fig. 3B? Is the knockout incomplete? And why is the level at 5 months so much higher than the ~20% level at only 4 months in the JCI paper?

This is likely a technical issue. Sometimes the confocal plane in WT neurons shows part of the neuron with nuclear NeuN staining but very little (thin) cytoplasm. Regarding the nKO samples, the knockout occurs only in *CamKII α* -positive neurons, leaving a population of *CamKII α* -negative neurons unaffected and *Cox+*. As discussed above, our subunit quantification results slightly varied from Peralta et al. due to differences in normalization methods. Moreover, western blots are prone to variability if performed in different conditions/different days. Figure 3B shows the quantification of proteins in cortex homogenates, which contain neuronal and non-neuronal cells. The fact that we analyzed samples at 5 months could have influenced the levels of *Ndufs3* or COX1, as there are more astrocytes at 5 months than at 4 months.

Referee #3

1) Throughout the manuscript, at times male and female data are combined and at other times they are separated. Based upon your data, sex appears to be a significant biological variable. Male/female data should be presented separately (different bars) but displayed in the same graphs in the main figures of the manuscript. Exact numbers of males and females included per cohort or experiment need to be indicated in figure legends.

In toto, our data suggests that sex is not a significant biological variable in these analyses. However, as requested, we have separated out the data (Fig.2). We have updated the figure legend to specify males/females per cohort.

2) Present doses in vg/kg.

Mice were injected with 2.8×10^{13} vg/kg for *Ndufs3*-nKO and 3.87×10^{13} vg/kg for *Cox10*-nKO mice. This has been included in Methods (Page 15, line 11-12).

3) Figure 2B - Asterisk markings are not clearly positioned - significance appears to be between treated and GFP-treated only - not between GFP-treated and WT. Males and females should be separated for all data presented. Exact number included per cohort needs to be indicated.

We have corrected and clarified the weight graphs to better show the statistical differences.

4) Figure 2E - As open field tests apparently show no significant differences between WT and affected mice at any age, this is not very interesting and could be supplemental data.

We agree with the Reviewer and have moved this data to supplemental figures.

5) Page 6 - 1st paragraph - It is an overstatement to say that *Ndufs*-nKO mice show a trend in increase in mtDNA copy numbers based upon the data presented. Also, gene replacement does not appear to have any significant effect on this.

We agree with the Reviewer and removed these data.

6) The mouse cohort numbers used throughout this manuscript are very low. Significant differences are the only ones that should be called out - there are many instances where things are described as elevated or increased when the data show no significant differences. This needs to be corrected throughout the manuscript - almost every figure. The data that show no significant differences between groups can be moved to supplemental data.

We agree with the Reviewer and have modified those instances accordingly. The numbers were not high because of a combination of factors, such as obtaining enough mice (at the same time) with the required multiple alleles for both treatment and control groups as well as the need to inject high titers of rAAV. Nonetheless, based on the robust effect we reached significance in the majority of the analyses. An unfortunate exception was females *Cox10*-nKO, as described below.

7) *Cox10*-nKO female treated/untreated data need to be provided.(also in 7 and S2).

Due to issues with our mouse colony (unrelated to the study), we lost some animals in the course of the experiment and were unable to obtain enough *Cox10*-nKO females needed to complete the 6-month-old full cohort. For this reason, we did not provide molecular data for *Cox10*-nKO

females. Nevertheless, we believe that the data we have with *Ndufs3* (males and females) and *Cox10* (males) are strong enough to support our conclusions.

8) Figure 6 - female data need to be provided and presented alongside the male data.

We agree with the Reviewer and have prepared a figure containing these data (*Ndufs3*-nKO), for an expanded view figure (Fig.EV2).

11) Based upon Figure 7 data it seems there may be a loss of NeuN+ cells over time in KO-GFP mouse brains. Unclear if this is true for both models or only one but loss of a cell type does sometimes result in altered AAV vg persistence in tissues.

Neuronal loss was not observed in the *Ndufs3*-nKO mice at 5 months. We did observe neuronal loss in the *Cox10* nKO males at 6 months. Neuronal loss could certainly play a role in the AAV persistence in brain. However, because we quantified GFP+ cells among NeuN+ cells, neuronal loss was accounted for.

Figure 8 should be supplemental data and should include images from both models.

We believe Figure 8 backs up one of the most important conclusions of this work, as it shows that a relatively small percentage of neurons expressing the rAAV transgene is sufficient to prevent the phenotype. We have moved several parts of the other figures to extended view, but we would like to keep Figure 8 as a main figure. Representative images were included to illustrate the quantification, and the full data sets are provided for both models.

13) Some supplemental figures and legends need editing. S1 title says female mice but the legend says male mice. S3 title only mentions one model not both.

We have corrected the Supplemental figures and legend errors identified.

14) Some discussion on what other cell types may have been transduced by this vector is needed. AAV-PHP.eB does transduce other cell types, however the synapsin promoter is neuronal specific (Kugler *et al*, 2003). In clinical settings, where multiple tissues are affected, broad transduction could be beneficial and a ubiquitous promoter could be used. In our work, the genes of interest, when fully knocked out, are embryonic lethal. By utilizing a cell specific knockout, we were able to address whether a neuron that is experiencing a severe OXPHOS complex deficiency can be rescued via gene replacement. We used the synapsin promoter to assure robust expression in neurons, the affected cells. As discussed, there may be neuroprotection either via mitochondrial transfer, the neural milia, stem cells, or other mechanisms. We hope to explore the role of mitochondrial transfer in the future, however this is beyond the scope of the current work. We did add this clarification to the Discussion (page 12, lines 27-32).

15) The discussion over-states the results in many cases as the treatment effects in females are not as significant or were not presented.

We have clarified that the data are based on both sexes for *Ndufs3* (Paragraph starting at line 18, page 5) and males for *Cox10* (page 6, line 20).

16) Some discussion needed for why are there differences between the treated and affected mice when there were no differences between WT and affected mice for many of the assessments presented.

Between treated and not treated mice, we found clear and significant differences for the key players in the pathogenesis, including: levels of the knocked out protein (NDUFS3 or COX1 [surrogate of COX10]), levels of the affected OXPHOS complexes, neuropathology and behavioral phenotypes. As most work with mice, there is some variability. Still, we believe this work provides solid evidence that the onset of the neuropathy could be prevented by using the AAV-PHP.eB vectors in the neuronal KO models. As an extreme example, we had a group of treated *Ndufs3* nKO mice sacrificed at 15 months, and they showed essentially no phenotype. We have never had a *Ndufs3*-nKO living beyond 7 months (usually they have overt phenotypes by 5 months and all dead by 6 months with massive neurodegeneration).

References-Response to Reviewers

- Aldridge GM, Podrebarac DM, Greenough WT, Weiler IJ (2008) The use of total protein stains as loading controls: an alternative to high-abundance single-protein controls in semi-quantitative immunoblotting. *J Neurosci Methods* 172: 250-254
- Dittmer A, Dittmer J (2006) Beta-actin is not a reliable loading control in Western blot analysis. *Electrophoresis* 27: 2844-2845
- Funfschilling U, Supplie LM, Mahad D, Boretius S, Saab AS, Edgar J, Brinkmann BG, Kassmann CM, Tzvetanova ID, Mobius W *et al* (2012) Glycolytic oligodendrocytes maintain myelin and long-term axonal integrity. *Nature* 485: 517-521
- Goasdoue K, Awabdy D, Bjorkman ST, Miller S (2016) Standard loading controls are not reliable for Western blot quantification across brain development or in pathological conditions. *Electrophoresis* 37: 630-634
- Kugler S, Kilic E, Bahr M (2003) Human synapsin 1 gene promoter confers highly neuron-specific long-term transgene expression from an adenoviral vector in the adult rat brain depending on the transduced area. *Gene Ther* 10: 337-347
- Takahashi S (2021) Neuroprotective Function of High Glycolytic Activity in Astrocytes: Common Roles in Stroke and Neurodegenerative Diseases. *Int J Mol Sci* 22
- Yardeni T, Eckhaus M, Morris HD, Huizing M, Hoogstraten-Miller S (2011) Retro-orbital injections in mice. *Lab Anim (NY)* 40: 155-160

2nd Jul 2024

Dear Prof. Moraes,

Thank you for the submission of your revised manuscript to EMBO Molecular Medicine. I am pleased to inform you that we will be able to accept your manuscript pending the following final amendments:

1) Please address all the points and implement all suggestions raised by the referee #3. Acceptance of the manuscript will depend on the completeness of your responses included in the next, final version of the manuscript. For this reason, and to save you from any frustrations in the end, I would strongly advise against returning an incomplete revision.

2) Figures: We note that the western blot loading control "total protein" is reused in several figures (Figure 3D-6C and Figure 4A,E-7E). Please indicate this clearly in the figure legends and make sure that the blots displayed in different figures are indeed from the same experiment.

3) In the main manuscript file, please do the following:

- Please address all comments suggested by our data editors listed below:

o Figure legends:

1. Please note that the exact p values are not provided in the legends of figures 2b, d; 3b-c, e-h; 4b-d, f, j; 5b, d, f; 6d; 7b, f-g; 8b; EV 1b-c, f-j, l.

2. Please note that in figures 2b, d; 4b-d, f, j; 5b, d, f; 7b, f-g; EV 2b; EV 3c, e; EV 5o; there is a mismatch between the annotated p values in the figure legend and the annotated p values in the figure file that should be corrected.

3. Please note that for the figures EV 4d-e; p-values and statistical tests are indicated in the legends. However, comparison for the same, "****/**/**/*" has not been represented in the figures. Please rectify this in the figures or legends as applicable.

- Add up to 5 keywords.

- Add callouts for Figure 8C and D.

- Rename "Conflict of Interest statement" to "Disclosure and competing interests statement". We updated our journal's competing interests policy in January 2022 and request authors to consider both actual and perceived competing interests. Please review the policy <https://www.embopress.org/competing-interests> and update your competing interests if necessary.

- Author contributions: Please remove it from the manuscript and specify author contributions in our submission system. CRediT has replaced the traditional author contributions section because it offers a systematic machine-readable author contributions format that allows for more effective research assessment. You are encouraged to use the free text boxes beneath each contributing author's name to add specific details on the author's contribution. More information is available in our guide to authors:

<https://www.embopress.org/page/journal/17574684/authorguide#authorshippinguidelines>

- We would encourage you to use 'Structured Methods', our new Methods format. According to this format, the Methods section should include a Reagents and Tools Table (listing key reagents, experimental models, software and relevant equipment and including their sources and relevant identifiers) followed by a Methods and Protocols section in which we encourage the authors to describe their methods using a step-by-step protocol format with bullet points, to facilitate the adoption of the methodologies across labs. More information on how to adhere to this format as well as downloadable templates (.docx) for the Reagents and Tools Table can be found in our author guidelines:

<https://www.embopress.org/page/journal/17574684/authorguide#structuredmethods>

<https://www.embopress.org/doi/full/10.1038/s44320-024-00037-6#sec-4>

- Indicate in legends number and nature of replicates and exact p= values, not a range, along with the statistical test used. To keep the figures "clear" some authors found providing an Appendix table Sx with all exact p-values preferable. You are welcome to do this if you want to.

- If no data are deposited in public repositories, please replace current text in data availability statement with the following sentence: This study includes no data deposited in external repositories.

Please check "Author Guidelines" for more information.

<https://www.embopress.org/page/journal/17574684/authorguide#availabilityofpublishedmaterial>

4) Appendix: Please add table of content with page numbers on the title page. The "Information of clinical cell lines" should be presented in a table format, named Appendix Table S1 and the table should be called out in the main manuscript text. Please also correct the nomenclature for the figures to "Appendix Figure S1" etc. and updated their callouts in the main manuscript text. Please delete the image of the movie file and the legend.

5) Funding: Please merge it with Acknowledgement and make sure that information about all sources of funding are complete in both our submission system and in the manuscript. Currently, project number for Stein Award is missing in the manuscript.

6) The Paper Explained: Please add it to the main manuscript text.

7) Synopsis:

- Synopsis image: Please incorporate short description into the image (tissue, cells, staining) and a color legend. Please upload the image as a high-resolution jpeg file 550 px-wide x (250-400)-px high. Alternatively, you may wish to provide a visual abstract to illustrate your article.

- Please check your synopsis text and image before submission with your revised manuscript. Please be aware that in the proof

stage minor corrections only are allowed (e.g., typos).

8) For more information: This space should be used to list relevant web links for further consultation by our readers. Could you identify some relevant ones and provide such information as well? Some examples are patient associations, relevant databases, OMIM/proteins/genes links, author's websites, etc...

9) Source Data: Please upload complete source data check list and all requested source data as one (zipped) file per figure for the main figures and group all source data for EV and Appendix figures in one zipped file.

10) As part of the EMBO Publications transparent editorial process initiative (see our Editorial at <http://embomolmed.embopress.org/content/2/9/329>), EMBO Molecular Medicine will publish online a Review Process File (RPF) to accompany accepted manuscripts. This file will be published in conjunction with your paper and will include the anonymous referee reports, your point-by-point response and all pertinent correspondence relating to the manuscript. Let us know whether you agree with the publication of the RPF and as here, if you want to remove or not any figures from it prior to publication. Please note that the Authors checklist will be published at the end of the RPF.

11) Please provide a point-by-point letter INCLUDING my comments as well as the reviewer's reports and your detailed responses (as Word file).

I look forward to reading a new revised version of your manuscript as soon as possible.

Yours sincerely,

Zeljko Durdevic

*** Instructions to submit your revised manuscript ***

1) a .docx formatted version of the manuscript text (including Figure legends and tables)

2) Separate figure files*

3) supplemental information as Expanded View and/or Appendix. Please carefully check the authors guidelines for formatting Expanded view and Appendix figures and tables at <https://www.embopress.org/page/journal/17574684/authorguide#expandedview>

4) a letter INCLUDING the reviewer's reports and your detailed responses to their comments (as Word file).

5) The paper explained: EMBO Molecular Medicine articles are accompanied by a summary of the articles to emphasize the major findings in the paper and their medical implications for the non-specialist reader. Please provide a draft summary of your article highlighting

6) For more information: There is space at the end of each article to list relevant web links for further consultation by our readers. Could you identify some relevant ones and provide such information as well? Some examples are patient associations, relevant databases, OMIM/proteins/genes links, author's websites, etc...

7) Author contributions: the contribution of every author must be detailed in a separate section.

8) EMBO Molecular Medicine now requires a complete author checklist (<https://www.embopress.org/page/journal/17574684/authorguide>) to be submitted with all revised manuscripts. Please use the checklist as guideline for the sort of information we need WITHIN the manuscript. The checklist should only be filled with page numbers where the information can be found. This is particularly important for animal reporting, antibody dilutions (missing) and exact values and n that should be indicated instead of a range.

9) Every published paper now includes a 'Synopsis' to further enhance discoverability. Synopses are displayed on the journal webpage and are freely accessible to all readers. They include a short stand first (maximum of 300 characters, including space) as well as 2-5 one sentence bullet points that summarise the paper. Please write the bullet points to summarise the key NEW findings. They should be designed to be complementary to the abstract - i.e. not repeat the same text. We encourage inclusion of key acronyms and quantitative information (maximum of 30 words / bullet point). Please use the passive voice. Please attach these in a separate file or send them by email, we will incorporate them accordingly.

You are also welcome to suggest a striking image or visual abstract to illustrate your article. If you do please provide a jpeg file 550 px-wide x 300-600px high.

10) A Conflict of Interest statement should be provided in the main text

11) Please note that we now mandate that all corresponding authors list an ORCID digital identifier. This takes <90 seconds to complete. We encourage all authors to supply an ORCID identifier, which will be linked to their name for unambiguous name identification.

Currently, our records indicate that the ORCID for your account is 0000-0002-8077-7092.

Link Not Available

12) Include a Reagents and Tools Table as part of the Methods section, which can be downloaded from our author guidelines (<https://www.embopress.org/page/journal/17574684/authorguide#structuredmethods>)

Photos 400-800 DPI

*Additional important information regarding figures and illustrations can be found at

<https://bit.ly/EMBOPressFigurePreparationGuideline>. See also figure legend preparation guidelines:

<https://www.embopress.org/page/journal/17574684/authorguide#figureformat>

***** Reviewer's comments *****

Referee #2 (Remarks for Author):

The authors have adequately addressed my points and I consider the manuscript now suitable for publication

Referee #3 (Remarks for Author):

This revised manuscript is improved but there remain a few points that should be addressed.

Figure 2 (and several other figures) - In general, it would make more sense to have each figure's story flow from left to right and then down as opposed to top to bottom and then back up to the top. Also, for figure 2, your reader is more likely to compare top of B to top of D (to see when weight phenotype - or corrected one - correlates to a functional deficit or correction) as your two mouse models are not directly compared in the analyses. In other words, try to keep Ndufs3 data together and COX10 data together in panels or label each graph separately with panel letters.

Figure 2B and Page 4 Line 15 - Why do the female COX10 KO AAV-COX10 treated mice weigh so much more than their WT controls? This divergence appears to be unique to females and thus indicates that sex is a biological variable and should be mentioned and discussed. The statement that the treated groups "continued to gain weight, comparable to the WT mice" is not accurate as your data indicates the treatment has a peculiar effect on female COX10 mice.

Figure 2C - Survival curves should be presented with male and female data separated. This is especially important for the COX10 cohort.

Figure 2D - statistics should be run between males and females in these graphs - then you can say more definitively whether there is or isn't a M/F difference in this assessment.

Figure 3 and onwards - When cropped images are used, the full blot images are typically provided as supplemental or appendix data - perhaps I missed this, but I do not see them.

Page 6, Line 18 - I understand the need to use COX1 as a surrogate for COX10, however, the wording of this section should be changed because COX10 levels are not directly measured. There may be COX10 expression present in neurons that was below your detection level based upon the COX1 surrogate - or not enough expressed to correct the COX1 localization phenotype. I understand your general point, but it isn't accurate to title this subsection "COX10 levels are increased" when that is not actually assessed.

Page 7, Line 9 - Was this actually COX10 staining or COX1? This needs to be accurately described.

ANSWER TO REVIEWERS

Reviewers 1 and 2 were satisfied with the previous responses and did not have further concerns.

Reviewer 3 suggestions are addressed below.

- 1) Figure 2 (and several other figures) - In general, it would make more sense to have each figure's story flow from left to right and then down as opposed to top to bottom and then back up to the top. Also, for figure 2, your reader is more likely to compare top of B to top of D (to see when weight phenotype - or corrected one - correlates to a functional deficit or correction) as your two mouse models are not directly compared in the analyses. In other words, try to keep *Ndufs3* data together and COX10 data together in panels or label each graph separately with panel letters.
We changed Figures 2 and 8 as requested. We believe the other figures have a good flow.
- 2) Figure 2B and Page 4 Line 15 - Why do the female COX10 KO AAV-COX10 treated mice weigh so much more than their WT controls? This divergence appears to be unique to females and thus indicates that sex is a biological variable and should be mentioned and discussed. The statement that the treated groups "continued to gain weight, comparable to the WT mice" is not accurate as your data indicates the treatment has a peculiar effect on female COX10 mice.
Most of the *Cox10* nKO females were indeed heavier than the WT females at early age. We do not have an explanation for this (other than mouse variability) as it is not a consistent finding with this line. What the data showed clearly is that both males and females with a *Cox10* neuronal KO start to lose weight at 4 months. This was also observed in our previous report (Diaz et al., 2012) for the *Cox10* nKO. A similar pattern was observed for the *Ndufs3* nKO (Fig. 2C). In contrast, AAV-PHP.eB-*Cox10* injected mice did not show weight loss (Fig. 2E). We changed the text to read: "On the other hand, after administration of the respective conditionally deleted genes, both OXPHOS-nKO models showed normal posture and no weight loss".
- 3) Figure 2C - Survival curves should be presented with male and female data separated. This is especially important for the COX10 cohort.
We separated by sex as requested (new Fig. 2B).
- 4) Figure 2D - statistics should be run between males and females in these graphs - then you can say more definitively whether there is or isn't a M/F difference in this assessment.
Statistics were run between male and females for each group. Only *COX10*-nKO+GFP animals showed a significant difference in RotaRod performance and only at 4 months of age (exact P values now provided in supplementary table S3). Therefore, most data points showed that males and females behaved similarly.
- 5) Figure 3 and onwards - When cropped images are used, the full blot images are typically provided as supplemental or appendix data - perhaps I missed this, but I do not see them.
Our apologies. The source data was compiled but not attached to the submission. These are now provided as "Source Data."
- 6) Page 6, Line 18 - I understand the need to use COX1 as a surrogate for COX10, however, the wording of this section should be changed because COX10 levels are not directly measured. There may be COX10 expression present in neurons that was below your detection level based upon the

COX1 surrogate - or not enough expressed to correct the COX1 localization phenotype. I understand your general point, but it isn't accurate to title this subsection "COX10 levels are increased" when that is not actually assessed.

We agree with the Reviewer and have corrected the title to read "COX1 levels"

- 7) Page 7, Line 9 - Was this actually COX10 staining or COX1? This needs to be accurately described.

We agree with the Reviewer and have corrected the manuscript to state "COX1 staining"

16th Jul 2024

Dear Prof. Moraes,

We are pleased to inform you that your manuscript is accepted for publication and is now being sent to our publisher to be included in the next available issue of EMBO Molecular Medicine.
